

**The role of pre-existing jointing on damage zone evolution and faulting style of thin**
**competent layers in mechanically stratified sequences: a case study from the Limestone Coal**
**Formation at Spireslack Surface Coal Mine.**
Billy J. Andrews[*], Zoe K. Shipton, Richard Lord, Lucy McKay
Department of Civil and Environmental Engineering, University of Strathclyde, Glasgow, G11XJ,
Scotland
*Correspondence to:* Billy J. Andrews (billy.andrews@strath.ac.uk)
**Abstract.** Fault and fracture networks play an important role in sub-surface fluid flow and can act to
enhance, retard or compartmentalise groundwater flow. In multi-layered sequences, the internal structure
and growth of faults is not only controlled by fault throw, but also the mechanical properties of lithologies
cut by the fault. This paper uses geological fieldwork, combined with fault and fracture mapping, to
investigate the internal structure and fault development of the mechanically stratified Limestone Coal
Formation and surrounding lithologies exposed at Spireslack Surface Coal Mine. We find that the
development of fault rock, and complexity of a fault zone is dependent on: a) whether a fault is self-
juxtaposed or cuts multiple lithologies; b) the presence and behaviour of shale, which can lead to
significant bed-rotation and the formation of fault-core lenses; and c) whether pre-existing weakness (e.g.
joints) are present at the time of faulting. Pre-existing joint networks in the McDonald Limestone, and
cleats in the McDonald Coal, influenced both fault growth and fluid flow within these lithologies.
**1 Introduction**
The mechanical properties, thickness, and interface properties of lithologies in a stratigraphic succession,
referred to as mechanical stratigraphy, combine to influence the deformation style of a rock mass (e.g.
Ferrill *et al.* (2017)). The effect of mechanical stratigraphy on faulting, in particular normal faulting, has
been studied for sand-shale sequences (e.g. van der Zee & Urai (2005); Schmatz *et al.* (2010)),
interbedded limestones and marls (e.g. Ferrill & Morris (2003), (2008); Long & Imber (2011); Ferrill *et*
*al.* (2012)), and ignimbrites (Soden and Shipton, 2013). The lithology being cut by the fault influences
fault dip: strands in competent layers have steeper dips than those in incompetent layers (Ferrill and
Morris, 2008). The ratio of competent to incompetent lithologies thus affects fault style and displacement
profiles (Ferrill et al., 2017; Ferrill and Morris, 2008). When incompetent layers dominate the sequence,
folding is commonly observed with thin competent beds displaying fault-related folding (Ferrill and
Morris, 2008; Lăpădat et al., 2017). The presence of incompetent lithologies also restricts fault growth
with strands terminating at incompetent beds. This leads to faults with high aspect ratios orientated





parallel to the strike of bedding (e.g. Nicol *et al.* (1996); Soliva & Benedicto (2005); Roche *et al.* (2013)).
In addition to mechanical stratigraphy, pre-existing weaknesses play an important role in the nucleation
and development of faulting (Crider and Peacock, 2004).The impact pre-existing weaknesses have on
fault growth depends on the orientation of a weakness relative to the growing fault and the stress ratio
(Lunn et al., 2008; Peacock, 2001). The presence of pre-existing weaknesses can also influence the
development of fault rock. For example, Soden & Shipton (2013) demonstrated that layer and joint
spacing in ignimbrites affected the aspect ratio of clasts found within the fault core. Of course, mechanical
stratigraphy itself influences the orientation of pre-existing weaknesses (Wilkins and Gross, 2002).
Fluvial-deltaic sequences are characterised by cyclical sequences of limestone, sandstone, siltstone, seat-
earth, shale, and coal (Thomas, 2013). The competent lithologies in the sequence (limestone and
sandstone) commonly contain joints. Coal has a unique, distinctive blocky texture due to the presence of
two roughly perpendicular fracture sets called cleats (Laubach et al., 1998). Cleats are ubiquitous in coals
as diagenesis takes place, and thus represent pre-existing weaknesses which may affect the location,
orientation and length of faults (e.g. Peacock (2001); Walsh *et al.* (2002)).
This study utilises exceptional exposures of the Limestone Coal Formation (LCF) exposed at Spireslack
Surface Coal Mine (SCM), Scotland, to investigate the effect of lithology and pre-existing structures on
the growth of strike-slip faults. Field photographs were used to map the key structures and kinematics at
a 1:1,000 scale. High resolution photomontages were then used to map faults and fractures and investigate
the interaction of faults and fractures with lithology and jointing. We find that faults cutting multiple
lithologies are thin (<0.3 m), display a complex deformation pattern, and locally branch entraining lenses
of sandstone. We also find that pre-existing joints and lithology strongly affect the growth and fluid flow
history of small offset, self-juxtaposed, faults.

## 2. Geological setting

The Midland Valley of Scotland (MVS) is a 90 km wide, 150 km long, ENE-trending basin that opened
during the late Devonian to Early Carboniferous in response to back-arc extension within the Laurussian
Plate (Leeder, 1982, 1988). This was followed by a period of thermal subsidence which continued
throughout Namurian and Westphalian times leading to the deposition and preservation of thick coal
measures across much of the UK (Leeder, 1982; Figure 1a).
The MVS is bound by two major tectonic lineaments - the Southern Upland Fault (SUF) to the south and
Highland Boundary Fault (HBF) to the north (Figure 1a) (Bluck, 1984). Carboniferous basins that have
axes oblique to the main trend of the MVS (e.g. Central Scottish Coalfield; Francis (1991)). These basins
can reach over 6 km in thickness (Dean et al., 2011) and are often obscured by Quaternary deposits. Faults





with associated, localised folding within the MVS have a complex history of reactivation caused by
sinistral strike-/oblique-slip during the Tournaisian and dextral strike-/oblique-slip during Viséan to
Westphalian times (Browne and Monro, 1987; Rippon et al., 1996; Ritchie et al., 2003; Underhill et al.,

68  2008).

**2.1 Spireslack SCM**
Spireslack SCM, next to the now abandoned coal mining village of Glenbuck, South Ayrshire, Scotland
(Figure 1a) provides an exceptional exposure of Carboniferous rocks in a 1 km long void (Figure 1b).  A
20°- 40° southerly dipping slope along bedding planes ends in a <130 m high working face. The
stratigraphy, comprises a continuous succession of Viséan to Namurian strata including a complete
section through the Limestone Coal Formation (LCF) (Figure 1c) (Ellen et al., 2016, 2019).  Bitumous
coal is found in cyclical fluvio-deltaic sequences that outcrop across much of the dip-slope and high wall,
bounded by the Upper and Lower Limestone Formations. The Lower Limestone Formation represents
more marine-influenced facies including extensive, fossil-rich limestone units (e.g. The McDonald
Limestone) (Davis, 1972). Above the LCF the Spireslack Sandstone comprises of one channelised, and
two tabular, sandstone beds (Ellen et al., 2019).
Offsetting this stratigraphy are several fault zones with shallow slip vectors and variably complex internal
structures. In addition to the faults, at least five Paleogene basaltic dykes are observed, which Leslie *et*
*al.* (2016) suggest intrude along pre-existing faults. The rocks exposed at Spireslack SCM are part of the
Southern Limb of the upright, WSW-ENE trending Muirkirk syncline that formed in response to mid- to
late- Carboniferous sinistral transpression (Davis, 1972; Leslie et al., 2016). Leslie *et al.* (2016) attribute
the faulting and folding observed at Spireslack SCM to this deformation, and have observed no evidence
of the later widespread dextral deformation (e.g. Underhill *et al.* (2008)).



**Figure 1: Location map: a) Map of UK coalfields (adapted from Donnelly (2006)) showing the location of Spireslack SCM and Structural features of the Midland Valley of Scotland; b) Regional geology of Spireslack open cast coal mine (after Ellen et al. (2019)); c) Regional stratigraphy of Spireslack SCM (after Ellen et al. (2019)).**





## 3. Methods

### 3.1. Field mapping

Geological mapping of the dip-slopes captured all units between the sandstones and shales below the McDonald Limestones and the sandstone bed above the Muirkirk 6' Coal. Mapping was undertaken at a 1:1,000 scale onto printed aerial photography from Bing (2017). All faults with greater than 0.2 m offset were recorded. Printed field photographs were used to collect more detailed observations at several key sites.

### 3.2 Analysis of fault and fracture networks

#### 3.2.1 Mapping procedure

Fault and fracture mapping was undertaken using two datasets: (i) a UAV derived photomontage of the McDonald Limestone bedding plane collected by Dave Healy of Aberdeen University; and (ii) an auto rectified photomontage of the high wall collected by the British Geological Survey. Interpretation areas were selected from the dip-slope and high wall for analysis to understand the geometrical and topological properties, and cross cutting relationships of fault strands and joint sets. Due to the instability of the highwall, there was very little access to the foot of the highwall and the interpretations are made principally on the photomontage. The interpretation areas were scaled in ArcGIS with mapping being undertaken by the lead author at a scale of 1:30 for the dip-slope and 1:50 for the high wall. Lineament mapping was undertaken by the same operator, at the same scale, to limit the effect of subjective bias on the data collected (Andrews et al., 2019; Scheiber et al., 2015).

#### 3.2.2 Network analysis

Fracture topology describes a fault or fracture network as a series of branches and nodes (e.g. Manzocchi (2002); Sanderson & Nixon (2015),(2018)). A branch is a fracture trace with a node at each end. Nodes can occur where a fracture terminates into rock (i-node), abuts against another fracture (y-node) or crosses another fracture (x-node). The proportion of different node types (i, y, and x) can then be plotted on a triangular diagram for the purposes of characterising and quantifying the connectivity of the network (Manzocchi, 2002; Sanderson and Nixon, 2015). In this work we interpret fault and fractures as orientation sets and report fracture/branch trace length (tl), 2D fracture intensity (I), and the percentage of connected branches (Pc).

Once the faults and fractures were interpreted (digitised as separate datasets), a visual assessment of the network was undertaken followed by network analysis using the open source ArcGIS toolbox NetworkGT (Nyberg et al., 2018) and the following workflow:



1. *Define sets:* Six 'Interpretation boxes' were added as shape files to the ArcGIS (three along the
dip-slope and three along the high wall) and the orientation of faults and the fractures within
them analysed. Length-weighted rose diagrams with 5° bin widths were used to interpret the
'orientation sets' in the network using NetworkGT (Nyberg et al., 2018). The digitised fault and
fracture data sets were then combined using the merge function in ArcGIS, and all three
investigated separately.
2. *Branch & Nodes:* The topology of the network was extracted using the 'Branch and Node' tool,
which splits the fracture trace poly-line file into individual branches, and assigns nodes as a
separate point-files (Nyberg et al., 2018). The resulting network was visually checked for errors
(e.g. incorrectly assigned nodes) and manually adjusted in ArcGIS to remove spurious nodes and
branches. Data were then exported to excel for further analysis.
3. *Network analysis:* For each network, the following data was extracted;
a. *Network connectivity:* For each dataset with the data not split into sets, the node and
branch proportions were assessed using a triangular diagram (c.f. Sanderson & Nixon
(2015)). The percentage of connected branches was then calculated using Equation 1.
$$P_c = \frac{(3N_y + 4N_x)}{(N_i + 3N_y + 4N_x)} \qquad \text{(Equation 1)}$$

b. *Trace length:* The trace length of digitised networks and sets within each sample area were
assessed using trace length distributions (Andrews et al., 2019; Priest and Hudson, 1981),
with the minimum, maximum, and median trace length values used to compare analysis.
c. *2D fracture intensity:* We compare the intensity of the networks and sets within the
network using 2D fracture intensity (Equation 2) ($P_{21}$; Dershowitz & Einstein (1988);
Rohrbaugh *et al.* (2002)).
$$P21 = \frac{\sum tl}{Area} \qquad \text{(Equation 2)}$$






# 4. Results

## 4.1 General fracture observations

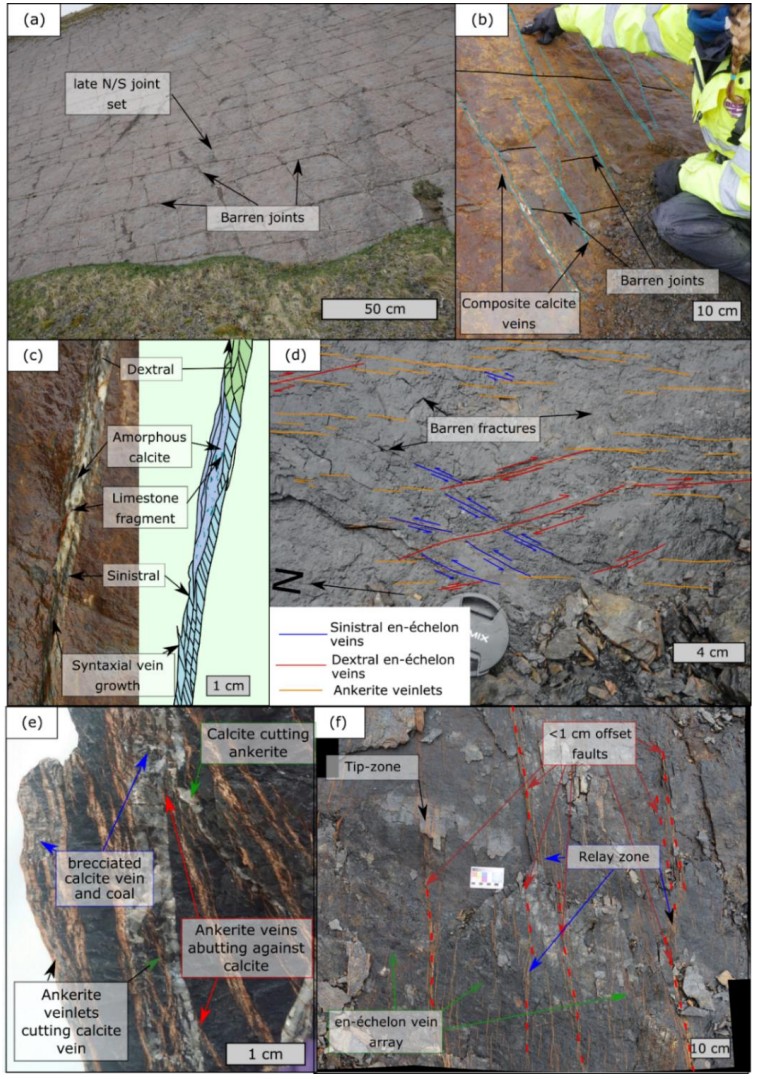

**Figure 2: Typical fracture properties for McDonald Limestone & McDonald Coal: a) barren joints observed away from faults across the southerly dipping (c. 40°) McDonald Limestone bedding plane; b) Mineralised N-S trending calcite veins, offsetting abutting E-W ladder joints on the bedding plane of the McDonald Limestone; c) annotated field photograph and interpretation of a multi-phase composite calcite vein exposed in the vicinity to a small offset fault along the McDonald Limestone Pavement; d) bedding plane exposure of mineralised fractures present within the Muirkirk 6' coal; e) annotated hand specimen displaying the vein relationships present during the faulting of the Muirkirk 6' coal; and f) the larger-scale mineralisation pattern as you move towards small offset faults in the Muirkirk 6' coal.**



Fractures at Spireslack SCM typically occur in two orthogonal directions that vary throughout the site
(NS and EW) and can be classified as either joints or shear fractures (often found in the proximity to
faults). Away from faults, joints in the McDonald Limestone form two orthogonal barren sets trending
roughly NE-SW and NNW-SSE. Orientation of these sets vary, with up to 20° of strike rotation observed
throughout the site. Cross cutting relationships show that there are multiple 'age sets' (Figure 2a). NE-
SW joints formed initially followed by sparsely spaced NE-SW joints then more NE-SW joints, which
abut against the pre-existing NNW-SSE trending joints. Finally, a dense network of N-S joints abuts
against both sets of E-W trending joints.
Calcite mineralisation is observed in the vicinity of, and along, primarily NW trending shear fractures
(Figure 2b). Mineralisation occurs as two styles: 1) amorphous, where no growth structures are present
and occasional fragments of limestone are observed within the mineralised zone, or 2) with syntaxial
growth textures suggesting both sinistral and dextral motion during mineralisation. Along fault planes
and within a few meters of faults, composite veins commonly occur, with multiple growth stages and
evidence of reactivation (Figure 2c).
Fractures in the coal layers are commonly filled with a buff to orange coloured mineralisation, identified
in the field as ankerite (iron rich carbonate) (Figure 2d-f). Fractures in coal occur as:
- *Coal cleats:* Ubiquitous in all coals. Spacing (typically <2 cm) is dependent on bed-thickness,
coal quality and the presence of clastic material (e.g. shale partings) (Laubach et al., 1998).
- *Mineralised shear fractures:* Typically 2 to 15 cm in length, but increase to >1 m long as apparent
shear offset increases. Fractures < 15 cm long abut against EW trending cleats, with trace length
restricted by cleat spacing. The thickness of planar ankerite veins increases as the length of the
fracture.
- *En-echelon arrays:* En-echelon ankerite veins display both sinistral and dextral motion (Figure
2d). Dextral arrays occur both simultaneously with, and later than, sinistral arrays.
- *Barren shear fractures:* In addition to the cleat network, fractures that abut against all other
fractures and are often curved, have trace lengths typically between 5 to 15 cm. These may
propagate from the tip of pre-existing mineralised shear fractures (Figure 2d).
A complex chronology of fractures is observed in the Muirkirk 6' coal. In Figure 2d dextral offset en-
echelon vein arrays (red) cross cut earlier sinistral sets (blue), with the former abutting against mineralised
shear fractures. Barren shear fractures then abut against both sets displaying a curvature indicative of a
dextral stress state. Abutting relationships suggest the barren shear fractures likely formed at the same
time as the dextral en-echelon vein array; however, they were not connected to a source of mineral rich



fluids. In Figure 2e, multiple phases of mineralisation and reactivation of veins can be observed. Veinlets
of ankerite both abut against, and cut through the calcite vein associated with a nearby small (<5 cm)
offset fault. Brecciation of coal and calcite is also observed, with undisrupted ankerite veinlets cutting
through the breccia. This requires a minimum of four stages of mineralisation/deformation:
1) Ankerite veinlets formed along the NS striking face-cleats.
2) Faulting leading to the development of coal breccia and calcite veining which either cut across
or abut against pre-existing structures.
3) Brecciation of the calcite vein and coal leading to the development of a chaotic fault breccia.
The breccia contains angular clasts of coal and calcite within an amorphous calcite matrix.
4) Finally, a return to ankerite mineralisation with dextral en-echelon arrays developed alongside
barren tip-damage zones.
These observations suggest that initial deformation and associated mineralisation occurred over a wide
zone of en-echelon arrays (Figure 2d), which was strongly influenced by the pre-existing cleat network
(Figure 2e). En-echelon arrays then began to interact leading to the development of localised mineralised
shear fractures (Figure 2f). As the trace length of the shear fracture increased, so did the thickness of the
zone leading to the formation of a dense array of small offset (<1 cm) strands which interacted through
the development of relay-zones. A later dextral stress state, demonstrated by reactivated features (Figure
2e), lead to another phase of en-echelon veins (Figure 2c), which also locally developed into mineralised
shear fractures.
The other lithologies display a strongly developed fracture stratigraphy (c.f. Laubach *et al.* (2009)). The
McDonald Seat Earth exposed in the western panel (Figure 3a) lacks a well-developed joint pattern.
Instead, shear-fractures are observed in relation to small offset strike-slip faults which cut the dip-slope
(Figure 4a,b). Fractures are only found in close proximity to fault strands either forming sub-parallel to
fault strands in the hanging wall block, or oblique to the fault strands in relay zones and fault tips. These
fractures commonly display small sinistral and dextral offsets (mm to cm) and are typically barren,
although occasionally showing pyrite along the fracture plane. Sandstones displayed bed-bound joint-sets
in a similar manner to the McDonald Limestone. However, there was limited bed-parallel exposure to
explore the age and orientation of sets in sandstone lithologies. Seat-earth in the high wall, in contrast to
the dip-slope, displays a well-developed bed-bound fracture network. This suggests that mine-related
stresses have may have caused deformation of these lithologies and the natural network has been altered
by both subsurface and surface mining activities.





## 4.2 Fault observations

### 4.2.1 Fault kinematics & Self-juxtaposed faults

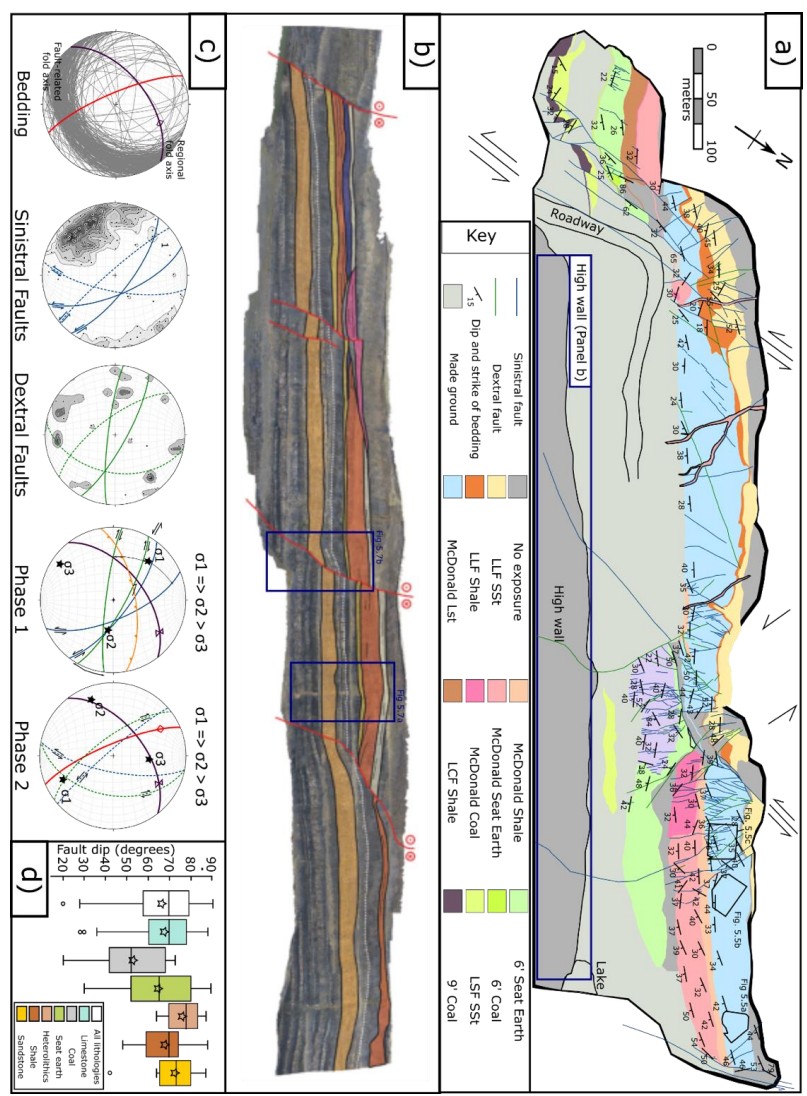

**Figure 3: Geological map of Spireslack SCM: a) Geological map undertaken as part of this study, displaying the locations of the detailed map-view fracture maps shown in Figure 5; b) Annotated photogrammetry of the high wall displaying the key stratigraphic horizons and faults (Ellen et al., 2019); c)) Fault kinematics by lithology. Stereographic projections were created using Stereonet 10.1 and contours represent 1% area; and d) box and whisker plots for fault dip by lithology.**




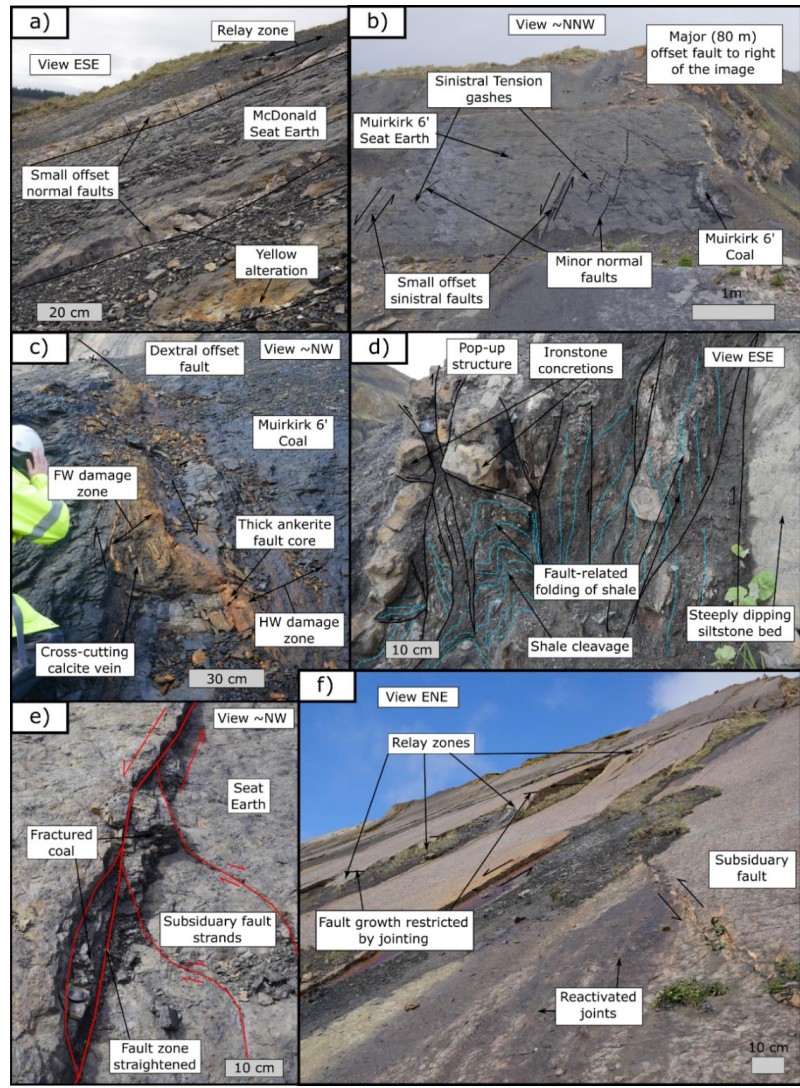

229

**Figure 4: Characteristic observations of Self Juxtaposed Faults (SJFs): a) Small-offset (c. 15 cm) fault strands and relay structures, and b) tension gashes and small offset normal faults exposed within the McDonald Seat Earth in seat-earth exposed to the far west of Spireslack SCM; c) symmetric damage zone and thick zone of ankerite mineralisation along a c. 40 cm offset dextral offset fault cutting the Muirkirk 6' Coal; d) bed-parallel thrusts and folding developed within the shale which underlies the McDonald Limestone to the NE of the site; e) the development of small pods of fractured McDonald Coal along a small offset sinistral fault exposed to the SW of the site; f) the interaction between faults and joints along the southerly dipping bedding plane of the McDonald Limestone.**

Several steeply dipping faults with low angle lineations (5° to 30°) were mapped at Spireslack SCM (Figure 3). Fault offset ranges from cm-scale, where displacement is limited to specific lithology (self-juxtaposed), up to the largest offset fault (c. 120 m true offset according to Ellen *et al.* (2016)) which cuts the east of the site. Most faults (75%) belong to a sinistral offset set, which formed simultaneously with





~NE trending dextral faults. Additionally, a later set of sinistral faults with offsets of cm's to m's, and
associated dextral faults, offset the earlier faults. Fault strike varies across the main void (Figure 3c), with
a N-S trend in the east and west of the site and a NW-SE trend in the centre. Fault dip depends on the
lithology cut by the fault. Dips in the McDonald Limestone range from 45° to 88° (mean = 69.1°, n = 47),
however, in coal seams fault dips range from 20° to 73° (mean = 49°, n = 24). In the shale interbeds, layer
bound bed-parallel thrusts (e.g. 040°/70° SE) with cm to m-scale offsets and associated folding can be
picked out where they offset ironstone layers (Figure 4d). The McDonald Seat Earth in the west of the
site displays dip-slip slickenfibers (50° to 60°), but only in faults with offset less than 1 m.
Self-Juxtaposed Faults, with offset less than 3 m, form either isolated strands (e.g. west of the void), or a
network of sinistral and dextral strands (e.g. near the centre of the void) (Figure 3). The internal structure
of self-juxtaposed faults depends on the lithology that the fault strand cuts (Figure 4, Table 1). Only large
offset fault strands can be traced between beds (e.g. the 5 m offset fault cutting the western panel; Figure
3), apart from where large packages of sandstone are found (e.g. the Spireslack Sandstone). For
lithologically restricted faults, trace length is typically low and well connected, with strands typically
abutting against another fault strand in < 15 m.
The majority of faulting at Spireslack SCM fits a sinistral-offset strain-ellipse (Figure 3c). In this model
the early dextral faults represent R' Riedel shears, with normal faulting of the McDonald Seat Earth,
thrusting in the shale and Riedel shears of the major fault strands which bound the workings developing
in the centre of the void. Bedding, which dips towards the south, matches a fold axis of 042°/80° N and
is likely to have been developed under the same stress state as the regional Muirkirk syncline. Faulting
that cuts the earlier structures (e.g. the oblique sinistral fault and minor dextral fault strands) does not fit
within this strain ellipse, and likely formed under a later dextral strain (Figure 3c). In addition to the two
phases of strike-slip tectonics, Paleogene dykes are observed exploiting pre-existing N-W trending fault
strands. These locally display pods of edge brecciation similar to that developed along faults in limestone,
and show dip-slip lineations suggesting there could have been a late stage of normal faulting.





| Lithology | Self-juxtaposed fault characteristics |
|---|---|
| McDonald Seat Earth | Segment linkage, folding, and increased fracturing between strands led to the development of a highly asymmetric damage zone (Figure 4a,f). Faults typically barren, only displaying yellow alteration and occasionally pyrite. |
| McDonald Limestone | Self-juxtaposed faults, associated relay zones, and nearby N-S trending joint sets, are mineralised (calcite), display high displacement to length ratios (2.4 to 2.8), and show extensive folding of the surrounding lithologies (Figure 5.4f). Strands often abut against favorably orientated pre-existing joints. |
| Coal | Fault strands are characterised by a fault core comprising of a 5 to 20 cm thick zone of ankerite, with occasional calcite mineralisation, brecciated coal and pyrite (Figure 4c). The fault core is discontinuous along strike, with displacement transferring to other strands after 1 to 5 meters (Figure 2c). The gentle folding of the bed between strands is taken up by a symmetric zone of damage consisting of increased fracturing, en-echelon veining and mineralised shear fractures. The structures represent a continuation of the processes discussed in Section 4.1.1. |
| Shale | Fault strands are rarely observed. High angle thrusts (40° to 60°) dominate, with bed parallel folding picked out by ironstone concretions (Figure 4d), which themselves can display internal deformation (tension gashes). Near self-juxtaposed faults a cleavage is developed sub-parallel to the fault plane, which combined with slickenfibers on competent bedding planes suggests bed-parallel slip. |

**Table 1: Self Juxtaposed Fault characteristics.**



## 4.2.2 Interaction between faults and fractures within the McDonald Limestone

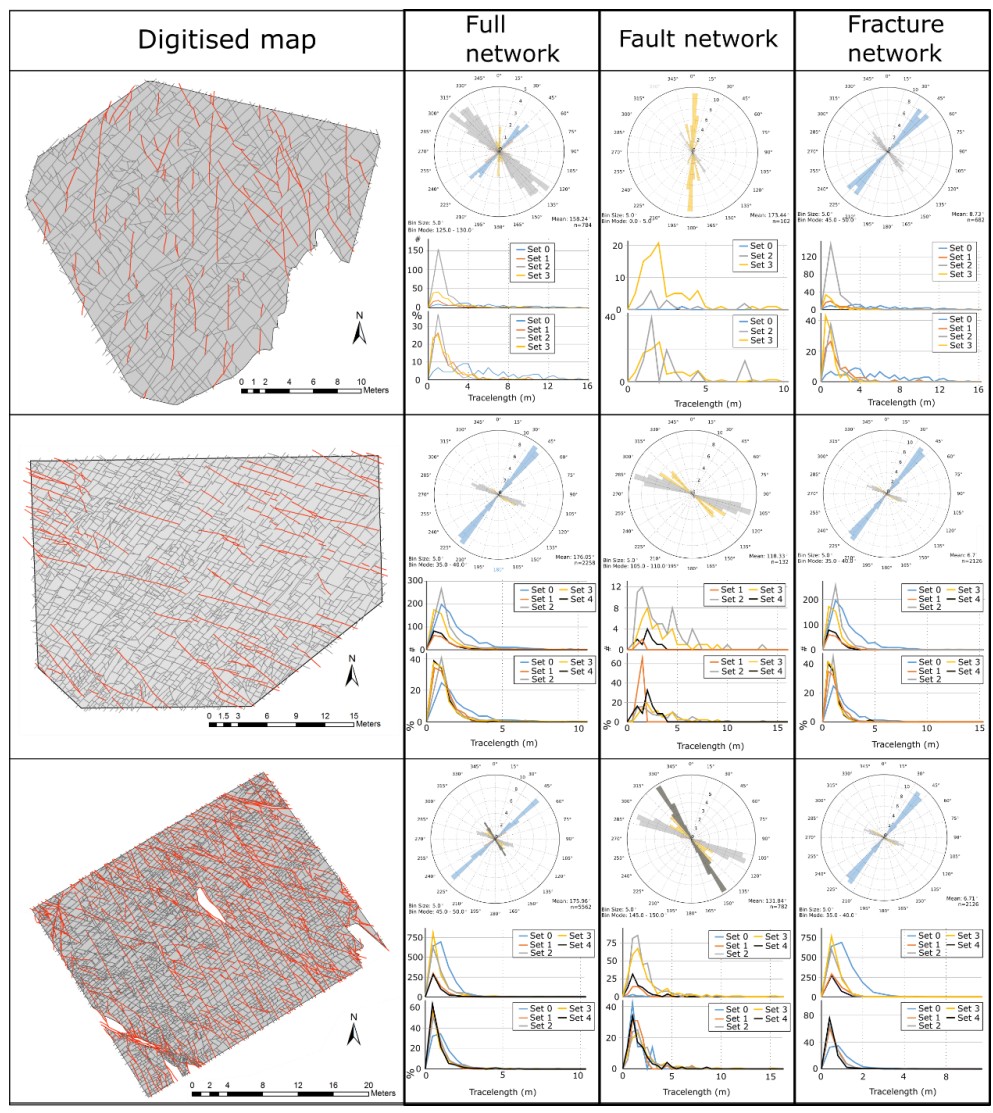

**Figure 5: Fracture maps with increasing intensity of faulting: For each digitised map the exported fault (red lines) and fracture (dark grey lines) maps, along with the interpretation areas used for the analysis (light grey) are provided. The orientation data, colour coded by sets, is then provided using length weighted rose diagrams with 5° bin widths. Trace length is presented as trace-length histograms as well as normalised trace-length histograms with bin widths of 0.25 m. Histograms are colour coded to match the sets outlined in the orientation data.**





The style of the fault and fracture network in the McDonald Limestone changes across the site (Figure 3)
with the chronology and network properties of each sample area described in Table 2. Overall, the
network is well connected and dominated by x- and y- nodes, with i-nodes only observed where faults
transfer displacement to another strand. As fault intensity increases, the complexity of age relationships
in the fault-fracture network increases. Where fault intensity is low and not favourably orientated to
reactivate joints (Figure 5), the age relationships match that described in Section 4.2.1. Across all sample
areas faults abut against the larger trace-length NE trending set (Figure 5), which are interpreted as
forming prior to faulting. When the interaction between faulting and jointing increases, either through an
increase in fault intensity or joints being favourably orientated for reactivation, age relationships become
complex with new fractures forming concurrently with faulting, probably during stress field rotation. The
fact that age relationships vary across the site suggests a highly heterogeneous stress field, which was
rotated relative to locally active fault strands. An increase in fault offset also affects the intensity, trace-
length and connectivity of the network.



| Sample area | Sets & age relations | Trace length & intensity characteristics | Network topology & connectivity |
|---|---|---|---|
| 1 Fig. 5a | Joint sets occur as two roughly perpendicular sets, an older 045° trending set and later 145° trending set. Faults are present as a separate NS trending set, which displace both joint sets, typically abut against large trace length NE trending joints, are mineralised and display sinistral syntaxial growth textures (e.g. Figure 2c). In the vicinity, and locally abutting against faults is a final stage of jointing, either associated with initial fault slip or later dextral reactivation. | The NE trending set ranges in has a larger trace length (4.10 ± 3.40 m) compared to the SE trending set (1.30 ± 1.10 m), with the latter typically abutting against the NE trending set. Trace lengths in the NS trending set range from 0.20 to 9.30 m (Median = 1.60 m), and typically abut against the NE trending set. Fault intensity is low (0.4 f/m), with moderate joint density (2.6 f/m) split into 1.3 f/m for the NE set, 1.0 f/m for the SE trending set and 0.3 f/m for obliquely aligned features. | The connectivity of the full network is high (Pc = 0.99) and is dominated by y nodes (76%), with X-nodes representing 23% of nodes. Field observations suggest the majority of x-nodes mapped on the drone map represent two y-nodes separated by <5 cm. The fault network is dominated by I nodes (90%) and is poorly connected (Pc = 24%). The joint network is dominated by y-nodes and has a connectivity of Pc = 95%. |
| 2 Fig. 5b | Dominated by barren joints, with faults displaying a NW trend, which reactivate appropriately orientated joints and abut against NE trending joints. The age relationships are complex and show multiple generations of joints, typically orientated in a NE or NW trend, however, many joints are observered which do not fit these sets. | Faulting is slightly higher intensity than SA1 (0.5 f/m), with fault trace length varying between 0.20 and 13.20 m (median 1.20 m). Small trace length faults are found oblique to the main strand (e.g. northerly trending faults median trace length = 0.80 m). Joint intensity is 3.1 f/m with the majority of fractures belong to the NE (1.7 f/m) or NW (0.7 f/m) trending sets. Fractures off this trend typically have smaller tl (Median = 0.60 to 0.70 m) compared to the NE (median = 1.30 m) and NW (median = 1.00 m) sets. | The connectivity of the full network is high (Pc = 0.99) and is dominated by y nodes (88%). The fault network is dominated by I nodes (89%) and is poorly connected (Pc = 28%). The joint network is dominated by y-nodes and has a connectivity of Pc = 90%. |
| 3 Fig. 5c | The complexity of joints varies considerably throughout SA3. Some areas display a simple relationship with an early ENE trending set and later NNW trending set, however, other areas display fracture corridors, which are aligned at a similar orientation to faulting, and which display multiple generations of joint formation. Faulting occurs as two sets, trending NNW, and NW. Faults typically abut against ENE trending joints, and locally cause the formation of new joints and rotation of pre-existing features. | Faulting intensity increases considerably in SA3 (1.9 f/m), with fault trace length ranging from 0.10 to 15.30 m (median = 1.40 m). The majority of faults trend between 125° and 155° (0.16 f/m) and display a higher median trace length (1.40 m) compared to other faults (1.10 m). Joint trace length is smaller in SA3 (0.50 m) compared to other sample areas (Median tl = 1.00 m and 0.80 m). Although most joint trend is NE (2.4 f/m) the NW trending set displays a wide range in orientation (125° to 155°) and 17% of joints are off axis from these trends. NE joints display a larger trace length, ranging from 0.00 to 5.50 m (median = 0.70 m) compared to other orientations (median = 0.40 m. | The connectivity of the full network is high (Pc = 0.99) and is dominated by y nodes (93%). The fault network is dominated by both I- (55%) and y-nodes (45%) and is moderately connected (Pc = 71%). The joint network is dominated by both i- and y-nodes and has a connectivity of Pc = 77%. |


**Table 2: Network characteristics for the sample areas outlined in Figure 5**


### 4.2.3 Large offset faults

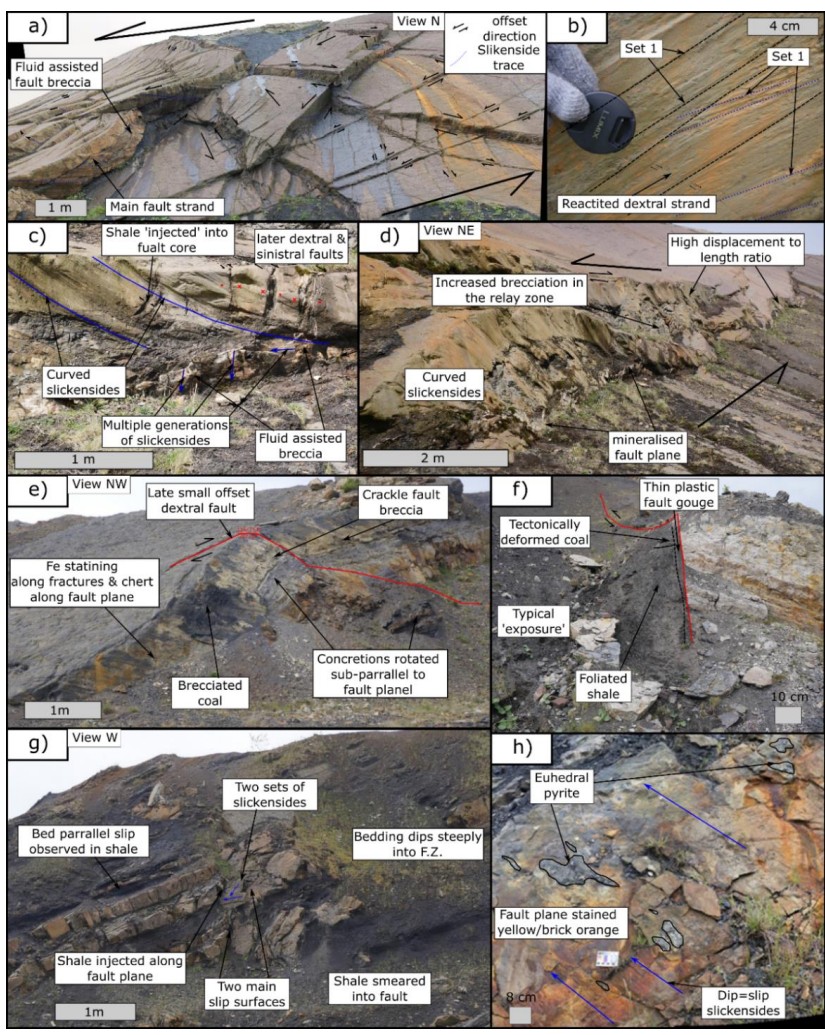

**Figure 6: Large offset fault characteristics: a) complex fault mesh consisting of multiple strands of sinistral and dextral strike slip fault planes (offset marked with arrows) picked out by shallow striations and the offset of the McDonald Limestone bedding plane; b) field photograph of a ~3 m offset fault strand within the complex mesh which displays multiple generations of fault striations, with local dextral reactivation separating striations belonging to set 2; c) fault architecture and d) view along strike of a 3 to 5 m offset fault strand exposed along the southerly dipping bedding dip-slope; fault architecture of the same 5 m offset fault cutting e) lithologies surrounding the McDonald Seat Earth, and g) interbedded sandstones, siltstones and shales of the Lower Limstone Coal Formation; f) primary slip plane of the ~80 m offset fault which cuts the west of the site; and h) shallowly dipping, sinistral dip-slip fault plane within a ~2 m thick sandstone bed of the Limestone Coal Formation.**





Faults that offset multiple lithologies (i.e. non self-juxtaposed) have a complex deformation style (Figure
6). Fault dips still vary depending on lithology, with steeper dips observed in competent lithologies for
the same fault. For example, a 4 m offset fault changes orientation from 135°/85° NE in the McDonald
Limestone to 110°/72° N in the McDonald Seat Earth. This change in orientation causes bed rotation and
the development of lenses, particularly in sandstones and seat-earths. Examples of larger-offset faults are
provided below, with the complexity of faulting depending on the lithologies cut by the fault (Figure 6)
and the plane of observation (i.e. map (Figure 3) vs high wall (Figure 7)).
*Example 1: Fault meshes in the McDonald Limestone and surrounding lithologies*
Faults cutting the McDonald Limestone with less than 3 m offset lead to the development of fault meshes
(Figure 6a). Rotation of bedding is accommodated along several fault-strands accompanied by the
development of tension gashes. The thickness of individual fault cores is low (<5 cm, Figure 6a,b), and
does not increase with displacement. Fault cores are mineralised, with local development of matrix-
supported breccias containing angular limestone clasts and clasts of re-worked calcite. These textures,
along with the development of Mode 1 fractures that offset previous slickenfibers (Figure 6c),
demonstrate fault reactivation. Multiple generations of slickenfibers are developed whose dip shallows
from the top to the base of the bed (Figure 6c, insert), providing further evidence of block rotation within
the fault zone. Folding and bed parallel deformation of the under- and over-lying shale helped
accommodate this rotation.
*Example 2: Dip-slip faulting of sandstones and seat earths*
3D exposures of faults cutting sandstone are rarely observed, however, in the center of the void there a 3
to 5 m offset fault cuts decimetre thick seat-earth and sandstones of the Limestone Coal Formation (Figure
3). The fault-plane is low-angled (100°/40° S) and displays dip-slip (40° to 55°) lineations. The fault plane
is altered to a brick-orange colour (Figure 6). Pyrite is locally preserved within corrugations along the
fault plane and consists of <4 cm euhedral crystals (usually <0.5 cm). The alteration and pyrite
preservation suggests sulphur-rich fluids migrated along the fault zone, and pods of crystal growth
developing elongated to the slip-vector suggesting this was syn-kinematic. Where coal is observed above
seat-earth (Figure 4e), brecciation of coal, thin zones of friable coal develop and cleats are rotated relative
to the orientation of the fault plane.





*Example 3: ~5 m offset fault cutting interbedded lithologies from the Lower Limestone Formation and*
*Limestone Coal Formation*
A ~5 m offset, sinistral fault is observed cutting limestones and sandstones of the Lower Limestone
Formation and the McDonald Seat-Earth to the west of the void (Figure 3). In the McDonald Seat Earth
(Figure 6b) fault dip changes from ~60° near the base of the outcrop to 007°/79° NE near the top and low
angle lineations (e.g. 20°/107°) and offset markers indicate a sinistral offset. The main fault plane is cut
by several later barren fractures (e.g. 116°/74° N and 292°/71° NE), which occasionally show cm-scale
sinistral offset (18°/019°). Brecciated McDonald coal is found within undulations on the fault plane. In
the underlying shale, several iron concretions (<10 cm) have been locally rotated and sheared in response to motion along the
fault. An asymmetric damage zone is developed, with minimal deformation of the footwall and a 20 to 30
cm wide zone of higher fracture intensity developing in the hanging wall. Bedding in the seat earth away
from the the fault displays gentle (2-5m wavelength), low amplitude (~50 cm) folding with wavelength
decreasing towards the fault.
In the underlying Lower Limestone Formation, the same fault develops a complex, 2 to 3 m thick,
mineralised fault zone (Figure 6c). The fault core is characterised by two mineralised slip surfaces
(216°/60° W & 261°/68° NW), each with shallow (10°/080°), moderate (25°/050°) and steeply (68°/083°)
dipping sets of slickenfibers developed. It is unclear which order these developed, and all apparently
display sinistral offset markers. Along the fault surface (015°/88° E), a ~5 cm thick pod of matrix
supported brecciated limestone is present in the hanging wall. Shale appears to have been locally injected
into fractures that had already been mineralised with calcite. To the north of the fault, the interbedded
sandstones, limestones, and shale dip steeply into the fault zone, reaching dips which match that of the
fault plane (60° to 70°). In contrast, bedding to the south displays only low amplitude folding (015°/56°
N; 043°/56° N).
*Example 4: 80 to 100 m offset fault cutting the full sequence*
The internal structure of the 80 to 100 m offset fault that cuts the west of the main void is only observed
at a single location (Figure 6e). The footwall comprises 6' Seat Earth that has been highly fractured,
juxtaposed against highly altered coal and folded shale with a steeply dipping cleavage. The fault core is
comprised of a thin (<5 cm), clay rich zone of plastic fault gouge containing <2 mm clasts of sandstone
and organic fragments. The altered coal has lost its cleat network and is noticeably harder than its
unaltered equivalent, creating a spark when struck with a geological hammer. This increase in coal rank
is potentially due to shear-heating (c.f. Fowler and Gayer, 1999; Li, 2001). Shear fractures in the
surrounding seat earth are often stratabound and increase in intensity towards minor-slip zones and the
fault core.


*Example 6: Fault strands cutting the high wall*

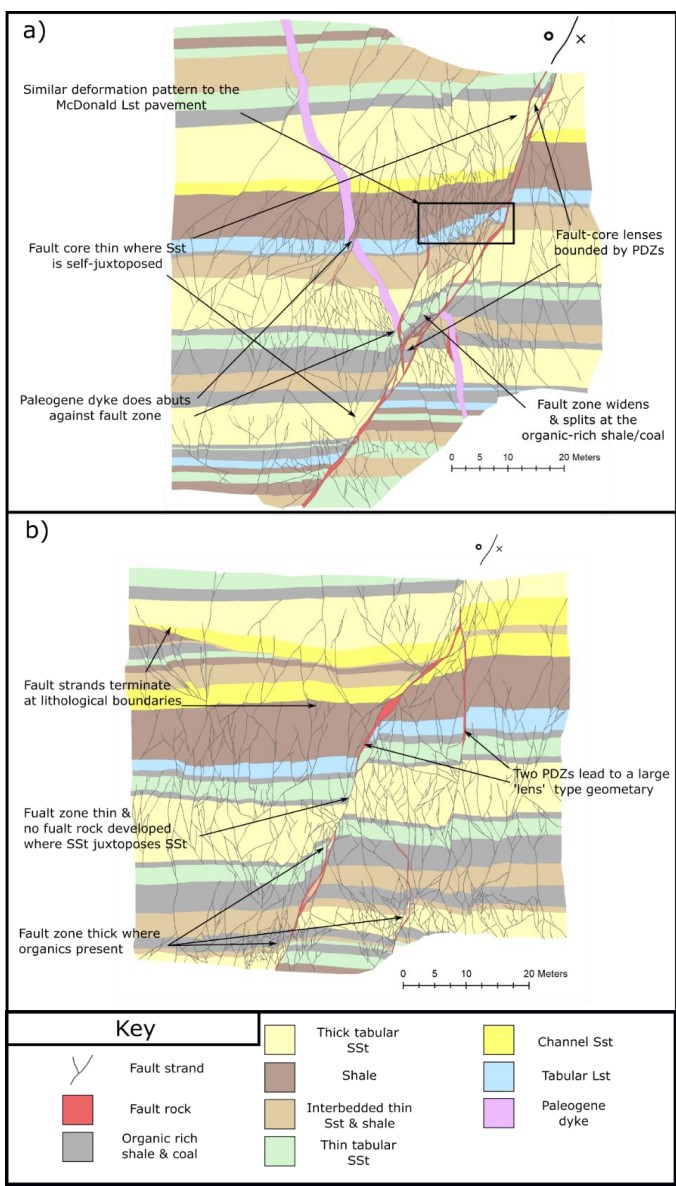


**Figure 7: Digitised fault strands of sinistral faults cutting the Limestone Coal Formation exposed along the high wall: a) sinistral fault which displays between 2 and 5 m of throw and has been cut by a later Paleogene dyke which is not observed with the main PDZ; b) sinistral fault with displays between 2 and 8 m throw along two PDZs.**



Fault strands cutting the high wall (Figure 7) appear to show a simpler geometry to those observed on the dip slope (Figure 6). It should be noted that because of the predominant strike-slip kinematics significant out of plane displacement exists, so visible offsets represent an underestimate of true displacement. The majority of throw is taken up by a small number of fault strands, particularly when faults cut channelised sandstones and limestones. Individual fault strands are thin and form an interconnected network of self-juxtaposed faults. Fault core thickness is typically below the width of a pixel on the orthorectified photographs (~5 cm), however, on the major faults the fault-rock thickness can be measured, although the rock type not quantified. The thickness varies considerably down-dip (<1.4 m), and while a continuous strand is observed in Figure 7a, in Figure 7b no fault rock is observed where the thick sandstone bed is self-juxtaposed.

The deformation style in the high wall varies depending on the lithological juxtaposition, with the proportion of sandstone in the faulted section controlling whether fault-core lenses are developed. For example, in both panels of Figure 7 fault-bounded lenses are seen in the lower third of the high wall. Faults are steep (apparent dip ~70° to 80°) with displacement taken up along a single fault strand, and damage zone evolution is low in areas where thick sandstone units are juxtaposed. However, where interbedded units are juxtaposed against each other the fault zone widens to 4 m in Figure 7a and 4.5 to 6 m in Figure 7b. Within these zones beds of competent lithology are rotated away from the main fault zone and subsidiary antithetic fault strands develop which abut against the main strand. Small offset faults are more abundant in the thick tabular sandstone, interbedded and shale units, with fault stands abutting and branching at lithologically controlled mechanical boundaries.

## 5. Discussion

### 5.1 The role of lithology on faulting style: self-juxtaposed vs non self-juxtaposed faulting

The observations at Spireslack SCM suggest that faults initiated both in the competent lithologies (sandstone, seat-earths and limestones) and the coals (Figures 3, 4). The properties of the faults (trace length, connectivity, D-L ratio and fault-rock development) depend on the lithology being cut, and the degree of non-self-juxtapostion. This is similar to observations of the growth of normal faults in interbedded limestones and marls (Ferrill et al., 2017), basalts (Ellen 2011), as well as 2D and 3D numerical modelling (Schöpfer et al., 2007, 2016). Self-juxtaposed faults developed in all the competent layers at the same time and initially grew as isolated strands, before interacting with other strands from the same unit. This behaviour matches established models for fault growth (Fossen and Rotevatn, 2016; Walsh et al., 2002; Wibberley et al., 2008).





However, large offset faults which breach more than a single lithology are strongly affected by the presence and behaviour of shale interbeds. Shale in the sequence behaves in a ductile manner with folds and cleavage developing (Figure 4d), and enables bed-parallel slip. In Figure 6g, shale is squeezed into pre-existing mineralised fractures, indicating the highly ductile nature of shale during faulting. The ductility of shale can be affected by many factors including lithology, mineral composition, organic carbon content, diagenesis, and thermal maturity (Wang and Gale, 2009). Burial depth is a major controlling factor for many of these properties and it is important to consider both the current and past burial depth (Yuan et al., 2017). As shale is buried and compressive stresses increases, the ratio of pre-consolidation stress and compaction-related stresses control the behaviour or shales and mud rocks (Yuan et al., 2017; Nygård et al., 2006). As a general rule, shales are ductile during burial, and brittle during exhumation where they experience stresses below the maximum stress they have encountered. While estimates vary across the Midland Valley, it is suggested the Limestone Coal Formation has a maximum burial depth of <3,000 m at around c. 60 Ma (Monaghan, 2014). Ductile behaviour of the shales at the time of faulting suggests that faulting was active during burial, rather than uplift. This will have enabled faults to initiate as isolated strands in competent lithologies. When faults accumulated enough displacement to cut multiple lithologies, shales accommodated the rotation of bedding, leading to rotated blocks and multiple generations of curved slickensides.

Self-juxtaposed faults coal remained relatively undeformed, or developed a thin zone of ankerite mineralisation. This differs from published examples in, where tectonically deformed coals in the form of soft-coal bands are often associated with normal faults (Godyń, 2016; Ju et al., 2012; Li et al., 2018), or bed-parallel slip in compressive environments (Frodsham and Gayer, 1999; Li, 2001). Soft coal bands often display a range of brittle and ductile features, for example S-C type cleavages, minor thrusts and folding (Li, 2001), all of which act to degrade the quality of the coal. Archival photographs of the far east of the site show that soft-coal bands were also not developed where coal was extracted from an area where bedding steepened to ~70° (Ellen et al., 2016; Leslie et al., 2016).

The fault core of the large offset faults often contain pods of coal present as un-mineralised chaotic fault breccia (Figure 6e). These deposits form in asperities along the fault zone, which get cut as the fault straightens. Asperities, formed by corrugations along the fault zone, have previously been identified both in the field (Sagy et al., 2007; Wright and Turner, 2006) and from seismic data (e.g. Lohr *et al.* (2008)). Asperities typically form aligned parallel to fault slip (Hancock and Barka, 1987), which is also observed in fault zones at Spireslack SCM. The behaviour of coal in the larger faults (Figure 6) differs from small offset faults (Figure 4) in that no mineralisation is observed. Where coal was observed overlying seat-earth (Figure 4e), coal was rotated, brecciated and thin zones of friable coal developed, suggesting that coal becomes entrained into the fault core as a rotated block, similar to a fault-core lenses (Gabrielsen et





al., 2016). This suggests that while self-juxtaposed faults can be used to understand fault growth up to a
certain point, once multiple lithologies are cut the processes change such that self-juxtaposed faults are
not representative of large offset faults (c.f. Ellen, 2011).

**5.2 Jointing and the effect of pre-existing weaknesses on the deformation style and fault growth**

The mechanically stratified succession at Spireslack SCM has led to the development of a fracture
stratigraphy (Laubach et al., 2009). Orthogonal joint sets are developed in the MacDonald Limestone
(Figure 2a), as cleats in the MacDonald coal (Figure 2d) and bed-bound joints within the sandstone layers.
While only two orientation sets are observed within the MacDonald limestone, abutting relationships
show these formed as at least 4 'age sets'. Similar observations reported for other sites (Peacock et al.,
2018; Sanderson, 2015) confirm that fractures in the same orientation, did in fact grow in response to
separate deformation events. Another way in which multiple age sets can develop is where the
intermediate ($\sigma$2) and minimum ($\sigma$3) principal stresses are nearly identical, and can therefore easily switch
between each other (Caputo, 1995; Caputo and Hancock, 1998). The ratio of principal stresses changes
the mechanical response of the layer (Healy et al., 2006; Moir, 2010; Moir et al., 2010)), with the dip and
dilatancy of fractures varying depending on the difference between ($\sigma$2) and ($\sigma$3) (Chang and Haimson,
2000; Haimson and Chang, 2000).
The joints at Spireslack SCM formed both prior to and associated with faulting: the sparsely spaced joint
set likely forming in response to far-field stress fields during burial, and later sets related to the early
stages of faulting and folding associated with the Muirkirk Syncline. This folding, and later faulting is
attributed to mid-to late Carboniferous sinistral transpression (Leslie et al., 2016). A late-stage dextral
event locally reactivates these structures, reactivates cleats within the coal (Figure 2 d, e, f), and locally
causes kink-bands to develop. This dextral strain was not identified in the work of Leslie *et al.* (2016).
and could be correlative to Upper Carboniferous deformation to the east of the MVS (Underhill et al.,
2008). Paleogene dykes, intruded along pre-existing NE to N trending faults, display a minor amount of
reactivation, with brecciation and dip-slip lineations developing along the margins. This suggests that late
stage extension, orientated to enable the reactivation of NE trending structures occurred since the
Paleogene, possibly linked to isostatic rebound or the opening of the North Sea or Irish Sea.
The presence of joints in the McDonald Limestone, and cleats within the Muirkirk 6' Coal influence the
internal structure and fault growth in these lithologies (Figure 4, 6). In both lithologies multiple sets of
pre-existing weaknesses existed at the time of faulting, however, it was only those orientated roughly
orthogonal to fault trend which caused fault strands to terminate (Figure 4). Coal cleats in the Muirkirk
6' Coal show evidence of reactivation (forming mineralised shear fractures and en-echelon arrays), and
may act to restrict the growth of these features. Although small-offset fault strands display evidence of



reactivation (e.g. brecciated coal, calcite and ankerite), further displacement is often taken up by the
formation of new shear fractures. Mineralisation of the cleats causes the strain-hardening of the coal with
pre-existing weaknesses (cleats) becoming mineralised strength inclusions. During the dextral
deformation stage new mineralised fractures formed, and tip-damage zones developed from the end of
shear fractures that had developed during the sinistral phase.
While joint sets in the McDonald Limestone may become rotated close to fault strands, no increase in
fracture intensity is observed and a typical core-damage zone structure is not developed (e.g. Caine *et al.*
(1996); Gudmundsson *et al.* (2010); Bense *et al.* (2013)). Mineralisation (primarily calcite) increases
towards the fault core, with fault cores in the McDonald limestone comprising of multiple generation of
slickensides, mineralisation and calcite matrix chaotic fault breccias (Figure 6a, c). While some rotation
of individual joints occurs towards the east of the site, the majority of joints remain planar and instead
acted as planes of weaknesses which became reactivated to accommodate fault slip. The rotation of joint
strike is in part related to the bulk rotation of competent beds along shale inter-beds into fault zones which
is observed both along the dip-slope and high wall. In Figure 5, the folding of the McDonald Limestone
can lead to previously mis-oriented joins becoming more favourably orientated to reactivation
Displacement on large offset faults, such as those observed in the high wall, is typically localised onto a
small number of principal displacement zones (Figure 7). This indicates that while jointing strongly
impacts early fault parameters, once a fault reaches a certain displacement, small-scale features such as
joints have only a minor effect on fault growth. The effect of joints on the early growth characteristics of
faults is discussed by Wilkins *et al.* (2001), who found faulted joints to develop little fault rock, and to
have considerably smaller displacement/length ratios that would be expected for faults which do not cut
jointed lithologies. Pre-existing joint-sets restrict fault-growth for self-juxtaposed faults through the
formation of faulted joints (Peacock, 2001; Soden et al., 2014; Wilkins et al., 2001), with lithology
becoming the major control once faults breach multiple layers (Nicol et al., 1996; Soliva and Benedicto,
2005; Wilkins and Gross, 2002). This behaviour is not observed in the McDonald Seat Earth, where
jointing is not present. Instead fault strands grow as single strands, which interact with other strands to
form tip-damage zones and relay zones where displacement transfers between fault strands (Fossen and
Rotevatn, 2016).

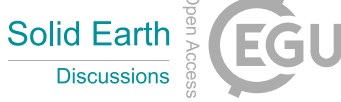

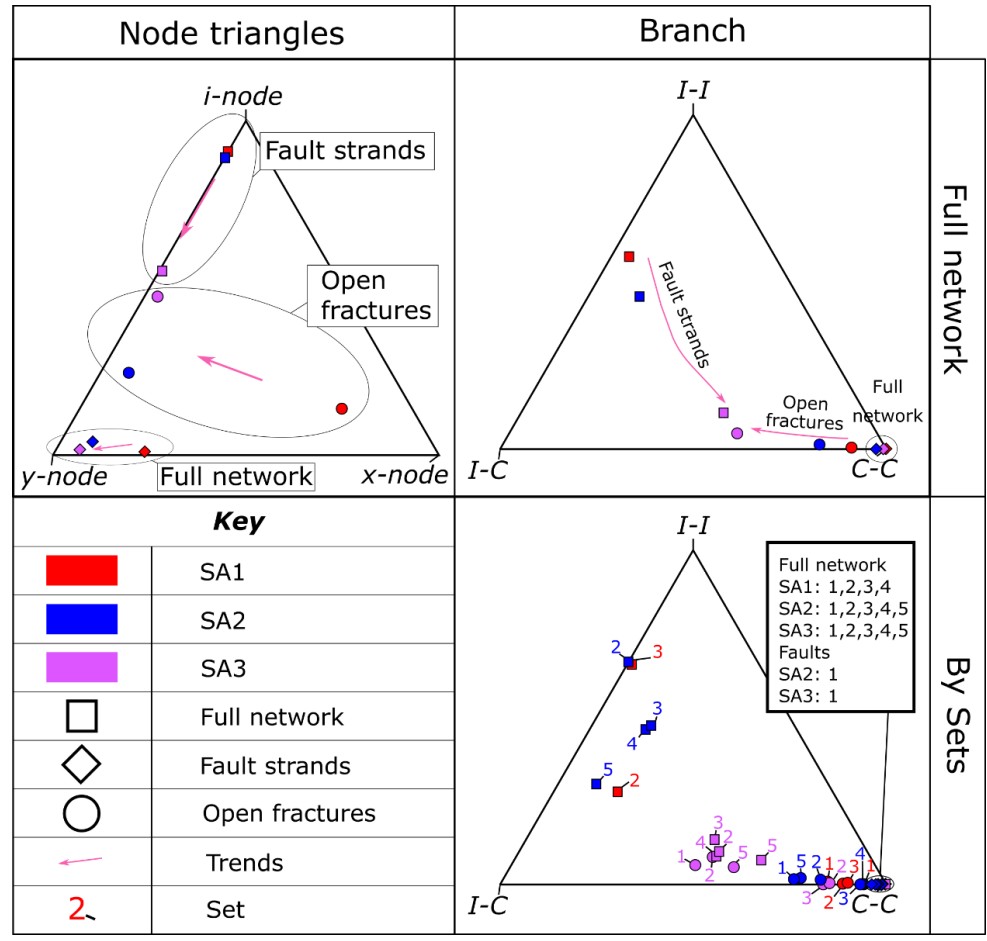

**Figure 8: Network topology data. Node and branch triangle (after Sanderson & Nixon (2015)) are presented for the full-network, mineralised fault strands, and open joints, for each of the three sample areas shown in Figure 5. Branch data is then presented by sets, as outlined in figure 5, to investigate the directionality of network connectivity.**





| | Network parameter | | Sample area | | |
|---|---|---|---|---|---|
| | | | SA1 | SA2 | SA3 |
| **a) Combined network** | **#lines** | | 784 | 2258 | 5562 |
| | **D (F/m2)** | | 3.1 | 3.5 | 5.9 |
| | **Pc** | | 1.00 | 0.99 | 0.96 |
| | **Tl (m)** | **Min** | 0.09 | 0.02 | 0.04 |
| | | **Max** | 14.71 | 13.16 | 15.33 |
| | | **Median** | 1.12 | 0.90 | 0.51 |
| **b) Fault network** | **#lines** | | 102 | 132 | 782 |
| | **D (F/m2)** | | 0.4 | 0.5 | 1.9 |
| | **Pc** | | 0.24 | 0.28 | 0.71 |
| | **Tl (m)** | **Min** | 0.22 | 0.21 | 0.10 |
| | | **Max** | 9.33 | 13.16 | 15.33 |
| | | **Median** | 1.62 | 2.34 | 1.36 |
| **c) Joint network** | **#lines** | | 682 | 2126 | 4778 |
| | **D (F/m2)** | | 2.6 | 3.1 | 3.9 |
| | **Pc** | | 0.96 | 0.90 | 0.77 |
| | **Tl (m)** | **Min** | 0.09 | 0.02 | 0.04 |
| | | **Max** | 14.71 | 10.33 | 5.49 |
| | | **Median** | 1.53 | 0.86 | 0.46 |

**Table 3: Overview of network properties for: a) the combined fault and fracture network; b) the**
**mineralised fault network; and c) the joint network which does not display mineralisation or**
**reactivation during faulting.**
Mineralisation along fault planes within coal (Figure 4), limestone (Figure 4 & 6), sandstone (Figure 6),
and to a lesser extent seat-earth (Figure 4 & 6), locally provides evidence of flow within the structures.
Fault-related veins display one or more crack-seal events (Figure 2c) indicating along-fault flow was
related to fault assisted opening of dilatational zones leading to the connection of pre-existing fractures
(Ferrill and Morris, 2003; Laubach et al., 2009; Ferrill et al., 2014). The multiple events suggest pathways
only remained open for a small amount of time and probably closed following fault slip (c.f. Sibson 1990,
1992). Faults in the McDonald Limestone behave in a similar way to other faults in carbonates with
primary slip surfaces becoming sealed following slip (e.g. Billi *et al.* (2003)).
Fault-related mineralisation in both limestones and coals would act to reduce connectivity, and hence
permeability, of the network (Figure 8). At the time of faulting, the majority of the network had been
formed, however, only joints/cleats orientated favourably for reactivation became mineralised. Following
mineralisation, these fractures became sealed and closed to future fluid flow. During faulting, the
connectivity of the network on the McDonald Limestone bedding plane varies depending on the intensity
of faulting (Table 3). Fault-assisted fluid flow in areas of low fault intensity (0.4 f/m$^2$; Figure 5) was
primarily confined to a sparse network of partially connected NS trending, poorly connected (Pc = 0.24)
fault strands. Where faulting of a similar intensity (0.5 f/m) is orientated favourably to reactivate joints





(SA2), the connectivity remains low (Pc = 0.28), however, fault trace length is greater due to the
orientation and spacing of pre-existing joints. When faulting intensity is high (1.9 f/m), the connectivity
of faults is high (Pc = 0.71), and both sets of joints become reactivated. Because multiple sets of joints
may restrict the growth of faults, trace length of individual fault strands is low and strain is taken up by
many small offset faults.
Faulting caused the development of several new joints, with joint intensity increasing from 2.6 in SA1
where limited shear fractures are observed, 3.9 f/m where fault intensity is high. The joint network initially
remains well connected (Pc = 0.96), however, as joints become reactivated connectivity drops to Pc =
0.90. In SA3, where fault intensity has increased to 1.9 f/m, the connectivity of the joint network drops
to Pc 0.77. It is also important to consider the orientation of the feature when considering fluid-flow
properties of the network. For example, while faults typically have a low to medium number of
connections per branch, those orientated between 060° and 100° plot close to the C-C vertex of the branch
triangle (Figure 8). This is also observed for joint sets, with those trending to the NW being the most
connected.
The evidence of transient fluid flow in both the McDonald Limestone and Muirkirk 6' Coal highlights
the importance of understanding the evolution of a fault and fracture network when assessing the
geological and fluid-flow history of a particular site. If fracture data were collected using the high-
resolution imagery alone, and not combined with field evidence, all fractures might be assumed to have
been open to flow. This would lead to a significant over-estimation of the permeability of the network.
For example, in SA3 the connectivity of all lineaments is Pc = 0.96, however, when only joints which
have not been reactivated by faulting are considered, this drops to Pc = 0.77. In this case the trace length,
which represents one of the most important parameters in fracture modelling (Lei et al., 2017; Min et al.,
2004) would also be overestimated, with mineralised fault strands displaying a larger median trace length
of 1.36 m. The presence of mineralisation only within particularly orientated joint or cleat sets also
highlights the importance of stress state on hydraulic properties of fractures (Cherubini et al., 2014).
While no data exist to quantify the magnitude of modern day stresses in Scotland (Comerford et al., 2018),
the stress orientations have been suggest as roughly EW extension (Baptie, 2010), and a northerly trending
maximum compressive stress (Heidbach et al., 2008). This stress orientation would tend to reduce the
aperture of large trace length ENE to NE trending joint sets, further reducing the modern-day connectivity
of the network.





**5.4 Implications for growth of strike slip faults**

While the role of mechanical stratigraphy on normal faults has received considerable attention, relatively few studies have focused on the effect on the development strike-slip faults (Gross et al., 1997; Nemser and Cowan, 2009; Sylvester, 1988). With the abundance of small-offset strike-slip faults in transtensional basins, it is of increasing importance to be able to predict the behaviour of such structures for hydrocarbon extraction (e.g. Gamson *et al.* (1993); Shuichang *et al.* (2009)), shallow geothermal projects (e.g. Malolepszy (2003)), carbon capture and storage (Solomon, 2007), and geotechnical engineering (Donnelly, 2006).

The amount of strike-slip on these faults is not quantified: the irregularity in the slip vectors makes calculations of total slip based on dip slip and slickenlines unreliable. However, the total slip is likely to be substantially greater than the observed dip-slip with slickenlines between 5 and 30 degrees rake, the total slip will be 2 to 11.5 times the dip-slip distance. The total thickness of all fault strands is low, with a maximum thickness of fault rock of 1.4 m (including fault-core lenses), and typically <30 cm. This value is low even for scaling of dip-slip displacement and thickness. This implies that the fault zone thickness has not grown as a function of the total slip.

The orientation of the pre-existing weaknesses and bedding with respect to the fault growth geometry is different for strike slip faults than normal faults. We find the growth of mineralised shear fractures and self-juxtaposed faults in coal to be retarded by the pre-existing joint network (Figure 2 & 4). Similarly, in Figure 5 favourably orientated joints either retarded the growth of self-juxtaposed faults, or are reactivated as shear fractures in the McDonald Limestone. The orientation of features that are orthogonal to propagation direction (e.g. NE joints cutting the McDonald Limestone), will cause a mechanical barrier. However, joints that are favourably orientated will be re-activated. Similar differences in mechanical response relative to an applied stress has been observed in rock deformation experiments for planar weaknesses (e.g. mud laminations (Whittles et al., 2002)) and cleats (Li et al., 2016). Unlike normal faults, where fault strands will step between joints in different beds (Wilkins et al., 2001; Wilkins and Gross, 2002), in strike slip faults self-juxtaposed faults will step in bedding-parallel view. The local orientation of joint sets, which is altered by the folding of competent layers (Figure 3) leads to the complex interaction of faults and joints observed at Spireslack SCM (Figure 5). In non-self-juxtaposed faults (Figures 6 & 7), it is instead primarily bedding which cause the termination of fault strands. Although this is similar to normal faults in mechanically layered sequences (Ferrill et al., 2017), the effect is less than would be expected and single through-going footwall strands are often observed. This suggests that the orientation of bedding and bed-perpendicular will have a significantly significant impact in the growth of strike slip faults than dip-slip faults.



## 6. Conclusions

The exceptional exposures of the Limestone Coal Formation exposed at Spireslack SCM enabled the effect of lithology and pre-existing structures on the internal structure, fluid flow properties, and growth faults to be investigated. We find that the internal structure of fault strands is strongly affected by: a) the lithology that was faulted; b) whether multiple lithologies are cut by the fault or not; c) the presence and behaviour of shale interbeds, and; d) the presence and orientation of pre-existing fractures. The geological evolution of Spireslack SCM displays a complex relationship of folding, brittle deformation and stages of mineralisation.

Faults in the McDonald Limestone and Muirkirk 6' Coal are strongly affected by the presence of the pre-existing joint and cleat network. In both cases, this causes the restriction of fault growth, with individual strands abutting against favourably orientated structures. The mineralisation of the cleat network in the Muirkirk 6' Coal led to an increase in the strength of the coal seam, with later reactivation concentrated at the tips of mineralised cleats. This is not observed in the McDonald Limestone, but the strength difference between the vein fill and the host rock is less. In both units, because fault planes become mineralised the permeability of the rock mass decreases as fault intensity increases. Our work demonstrates the importance of considering not just the lithologies being faulted, but also whether pre-existing weaknesses are present. Where pre-existing weaknesses are present, fault-growth will be restricted and the connectivity of a network can drastically change through time following mineralisation and/or changes in stress directions/folding.

## Acknowledgements

This work was funded through BJA PhD studentship, supported by the Environmental and Physical Sciences Research Council (EPSRC, award number EP/L016680/1). LMcK is supported by a University of Strathclyde Environmental and Physical Science Research Council (EPSRC) Doctoral Training Partnership (DTP) award (award reference 1904102). We would like to thank Dave Healy for the use of the high-resolution photomontage of the McDonald Limestone dip slope and the British Geological Society for the use of the photomontage of the high wall.



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
