# Peer review of "The growth of faults and fracture networks in a mechanically"

_Solid Earth, 2019_

## Referee Comment (RC1) · David Sanderson (Referee) · 14 Feb 2020

**Review of paper submitted to Solid Earth**

**Title:** *The role of pre-existing jointing on damage zone evolution and faulting 1 style of thin 2 competent layers in mechanically stratified sequences: a case study from the Limestone Coal 3 Formation at Spireslack Surface Coal Mine.*
**Authors:** Billy J. Andrews, Zoe K. Shipton, Richard Lord, Lucy McKay

This paper provides a detailed description of a very interesting exposure in central Scotland. The locality is a disused surface coal mine and contains some fractures and faults related to the evolution of this part of Scotland. More importantly, the exposure allows a detailed analysis of the relationship between different types of fracture – faults, joints and veins, and how these vary in a mechanically layered sequence. The paper uses a range of geometrical and topographical analyses to provide a good analysis of the resulting fracture networks. The importance of the site is clear from the paper and as such **the paper provides a very valuable study that should be published.**

**There are, however, a number of revisions necessary before the manuscript is acceptable; these include the following:**

**1. Tidy up the text by removal of inconsistencies and more precise definition of many of the terms used.**

I have added lots of specific comments and edits on the pdf. Most importantly many terms are used without clear definition, and their usage seems to vary from one section to another.

The use of 'self-juxtaposed fault' is particularly unclear. I think we need a clear definition of this term - the usage only becomes obvious as one reads the paper. In effect "self-juxtaposed" is really being used in place of "small". "Self-juxtaposed" describes the relationship between the wall-rock stratigraphy across a fault, and is thus a "topological" term describing the wall rocks NOT the fault, i.e. a small fault can produce a self-juxtaposition of the wall rock stratigraphy.

**2. A better presentation of the maps and data.**

Although this aspect of the paper is generally good, the diagrams are rather detailed and "formal" in their approach. There are many opportunities to provide a more visual presentation of the relationships between the data. For example, the site could be more clearly described by combining the maps in Figs 1 and 3 as follows:

[Figure]

This allows comparison of the observed detail with the surrounding faults mapped by BGS.

**3. The sequence of development of the structures.**

This is presented at four main parts of the paper: (a) a discussion of the mineralisation (Section 4.1, especially lines 194-200); (b) discussion of faults (mainly section 4.2.1); (c) interactions between faults and fractures (section 4.2.2); (d) large offset faults (Section 4.2.3). The material in these sections contains a lot of detailed information and it is difficult to relate much of this to the evaluation of the fracture sequence. In part this is caused by a somewhat inconsistent terminology, with the definition and criteria for recognition of the different fracture types being unclear. I would recommend having a separate section that clearly discussed the fracture history.

In Section 4.1 the authors recognise 4 types of fracture: coal cleats, en-echelon arrays, mineralised shear fractures and barren shear fractures. These are not clearly defined and their illustration in Fig. 2 introduces a number of different terms (joint, vein, etc.).

In Section 4.2.2 there is also a lot of discussion of "lineations", which appear to be mainly of low pitch, suggesting strike-slip. The authors also mention "reactivation" of the faults. I did not get a clear picture of how they view the faults. To me these could well be early normal faults (well described from elsewhere in central Scotland since the work of EM Anderson) that were later reactivated by strike-slip (again there are good example of strike-slip faulting elsewhere in the region). Is that the view of the authors? This should be discussed more clearly.

The authors use abutting and cross-cutting relationships to suggest relative ages of the different fractures. This generally works well for joints (especially where cross-cutting is rare) and veins, where cross-cutting can be used to determine relative age). The problem is that faults have displacement, therefore abutting does not work so simply and needs to take account of this displacement. The fact that many of the fractures are mineralised, should allow greater use of cross-cutting.

In section 4.2.2, the interaction of faults and fractures is discussed, mainly in terms of the 'joints'. A number of key questions largely remain unanswered (or if they are they are buried in the text and not obvious to me). How are these 'joints' distinguished from 'barren shear fractures'? Do shear fractures and faults clearly displace the joints. Are there cross-cutting relationships between veins and joints? If so what are the relative ages?

This summary of joint/fault relationships shown in Fig 5 (lines 276-288 and Table 2) is interesting. The table has a lot of very long sentences and is difficult to read. The single sentences are really a collection of phrases that could easily be separated (and arranged more logically). There also appears to be a lot of missing words.

The analysis contains a number of rather surprising conclusions:
1) the "joints" predate the "faults", yet the faults are mineralized!
2) How do the faults and joints fit into the sequence of mineralization (lines 194-200)?
3) One set of "faults develops sub-parallel to the c.N110E joint set - can these be described as "faulted joints" (in sense of Zhao and Johnson (1992, JSG 14, 225-36).

These are important outputs for this paper and need to be clarified and highlighted, and discussed later in the paper.

The abutting relationships in Fig 5 appear to contain many contradictions (although I agree that the faults look late). The problem is that abutting and cross-cutting work well for joints (especially where cross-cutting is rare) and veins, where cross-cutting can be used to determine relative age. The problem is that faults have displacement, therefore abutting does not work so well.

The section on larger faults (4.2.3) contains a lot of detailed observation, but I would like to have seen some synthesis of these details. The discussion is not helped by the repeated use of 'offset' instead of 'separation'. Given that 'offset' can mean either a displacement component or a separation, it would be best to abandon the term altogether and talk about separation (where only relationships of layers is known) and displacements (where kinematics is supported by striations and/or piercing points).

**4. Fault and fracture topology**

The topology of the different fracture types contains some clear errors. Since the procedures are not explained clearly enough, I am left to speculate as to the causes.

I think the analysis using NetworkGT was carried out as follows:

1. The entire network was digitized.
2. The data were divided into sets based on orientation and fracture type (faults and joints)
3. The nodes and branches for entire network were then calculated (using the tool in NetworkGT)
4. This would then give the correct results for the entire network. This looks to plot correctly on Fig. 8, as the diamond symbol, although these are erroneously assigned to the "fault strands" in the key. The correct assignment would be: squares – faults; circles - open fractures and diamonds – all network, as indicated on the node triangle.. The resulting values for Pc in Table 3 also appear to be correct (i.e. $0.96 \leq Pc \leq 1$).

To get the corresponding data for the faults and joints, the above steps should have been repeated for each fracture type. This is clearly not what was done, since the joints would have given very similar results to the entire network, i.e. high Pc and Y-node dominated. There are certainly not as high a proportion of I and X nodes as plotted in Fig 8. The results for the branch plots may be more robust.

I think that the nodes for the entire network were somehow distributed into values for the joints and faults, probably by removal of the fault-related nodes and assigning those that remain to the joints. This does not work because the majority of the nodes are produced by intersection with joints. I think there may be similar problems with splitting the fracture types into sets, as individual sets of sub-parallel fractures contain few connected nodes and very few I-I branches (c.f. as plotted).

This means that the discussion of the implications of flow based on Fig 8 and Table 3 is flawed.

**5. Network properties and flow/permeability**

There is also another major flaw in the discussion of flow and permeability in relation to the network characteristics. The reason this is flawed is that the network topology, essentially evaluates the connectivity of the fractures, whereas the permeability of the rock mass is a step-like function dependent on the percolation threshold, and, once this is reached, primarily on the conductance of the fractures on the connected component of the network. The correct topological analysis of each fracture type would allow discussion of this, but it would need to recognise the difference between the rolls of: (a) a conductive joint network, (b) the palaeo-flow in an active fault network, and (c) the subsequent effect on mineralization of the fault network. For example, superposition of a connected network of sealing faults would counteract the conductivity of an earlier network of open fractures.

This whole section talks about "open fractures". These are not clearly defined. How do they correspond to the 'barren shear fractures' and 'joints' discussed earlier.

**6. Abstract and conclusions**

These do a very poor job in summarising the content of the paper. There is too much "discussion" and arm waving about role of lithology, strengthening by mineralization, and effects on permeability, with not enough on the key issued of the sequence of development and interaction of the different fracture elements.

**7. Detailed comments are made on the annotated pdf**

David Sanderson

February 2020

[revised manuscript text omitted]

---

## Referee Comment (RC2) · Bailey Lathrop (Referee) · 27 Feb 2020

See attached PDF

[Figure]

**The role of pre-existing jointing on damage zone evolution and faulting style of thin competent layers in mechanically stratified sequences: a case study from the Limestone Coal Formation at Spireslack Surface Coal Mine.**
Billy J. Andrews*, Zoe K. Shipton, Richard Lord, Lucy McKay

Dear Editor,

This manuscript describes how mechanical stratigraphy and pre-existing weaknesses affect the style of strike-slip faulting in the Spiresack Surface Coal Mine, Scotland. The authors gave a history of the complex stages of folding and faulting in the Spirestack Coal Mine. They found that faulting in the study area was affected by the presence and orientation of pre-existing structures and whether the fault cuts through multiple lithologies, especially shale. This has implications specifically in the geologic understanding of the area, and generally for fluid flow and strike-slip fault kinematics.

This research has the potential to make a good contribution to Solid Earth, but might benefit from some revisions. I find few issues with science, but the writing and organization of the paper requires some thought. Overall the quality of writing could be improved, and an intense proof-read is likely required in order to make the paper more impactful. In no particular order, my main comments include:

- The writing style lacks clarity and is often difficult to read and understand
- The results section should be reorganised for clarity, and interpretations and results should be separated into different sections
- The paper needs a thorough proofread
- The figures are too busy, and could be simplified
- Relevant figures need to be referred to in the text more often
- The abstract and conclusions should contain more specific results

This paper will make a strong contribution to the field of strike-slip fault growth, and will be of interest to the readers of Solid Earth. I look at some similar research questions in normal faults, so I find the topic particularly interesting. The quality of writing could be improved throughout, and I have made some suggestions throughout the paper on how this could be done.

I am listing my main concerns with the paper below, and have made comments in a PDF of the paper. I would be happy to (and look forward to) reviewing this paper again after these points are addressed.

Kind regards,
Bailey Lathrop
Imperial College London

**Fig. 1.** Cover Letter

[Figure]

[Figure]

[Figure]

**The role of pre-existing jointing on damage zone evolution and faulting style of thin**
**competent layers in mechanically stratified sequences: a case study from the Limestone Coal**
**Formation at Spireslack Surface Coal Mine.**

Billy J. Andrews[a], Zoe K. Shipton, Richard Lord, Lucy McKay
Department of Civil and Environmental Engineering, University of Strathclyde, Glasgow, G11XJ,
Scotland
*Correspondence to:* Billy J. Andrews (billy.andrews@strath.ac.uk)

**Abstract.** Fault and fracture networks play an important role in sub-surface fluid flow and can act to
enhance, retard or compartmentalise groundwater flow. In multi-layered sequences, the internal structure
and growth of faults is not only controlled by fault throw, but also the mechanical properties of lithologies
cut by the fault. This paper uses geological fieldwork, combined with fault and fracture mapping, to
investigate the internal structure and fault development of the mechanically stratified Limestone Coal
Formation and surrounding lithologies exposed at Spireslack Surface Coal Mine. We find that the
development of fault rock, and complexity of a fault zone is dependent on: a) whether a fault is self-
juxtaposed or cuts multiple lithologies; b) the presence and behaviour of shale, which can lead to
significant bed-rotation and the formation of fault-core lenses; and c) whether pre-existing weakness (e.g.
joints) are present at the time of faulting. Pre-existing joint networks in the McDonald Limestone, and
cleats in the McDonald Coal, influenced both fault growth and fluid flow within these lithologies.

**1 Introduction**

The mechanical properties, thickness, and interface properties of lithologies in a stratigraphic succession,
referred to as mechanical stratigraphy, combine to influence the deformation style of a rock mass (e.g.
Ferrill *et al.* (2017)). The effect of mechanical stratigraphy on faulting, in particular normal faulting, has
been studied for sand-shale sequences (e.g. van der Zee & Urai (2005); Schmatz *et al.* (2010)),
interbedded limestones and marls (e.g. Ferrill & Morris (2003), (2008); Long & Imber (2011); Ferrill *et*
*al.* (2012)), and ignimbrites (Soden and Shipton, 2013). The lithology being cut by the fault influences
fault dip: strands in competent layers have steeper dips than those in incompetent layers (Ferrill and
Morris, 2008). The ratio of competent to incompetent lithologies thus affects fault style and displacement
profiles (Ferrill et al., 2017; Ferrill and Morris, 2008). When incompetent layers dominate the sequence,
folding is commonly observed with thin competent beds displaying fault-related folding (Ferrill and
Morris, 2008; Lăpădat et al., 2017). The presence of incompetent lithologies also restricts fault growth
with strands terminating at incompetent beds. This leads to faults with high aspect ratios orientated

[Figure]

**Fig. 2.** PDF With Corrections

---

## Referee Comment (RC3) · Bailey Lathrop (Referee) · 27 Feb 2020

2005.

[referee-annotated manuscript omitted]

---

## Author Comment (AC1) · 4 Jul 2020

Dear David Sanderson,

We would like to thank you for your in depth review of our manuscript 'The role of pre-existing jointing on damage zone evolution and faulting style of thin competent layers in mechanically stratified sequences: a case study from the Limestone Coal Formation at Spireslack Surface Coal Mine'. Please find attached our response to your comments as

well as the response to reviewer #2. The suggestions and comments have helped us greatly improve the structure and clarity of our work and we hope that the restructure of the manuscript now means it can be accepted for publication in Solid Earth. In particular we would like to thank you for pointing out the requirement for a robust order of events, something that we agree was lacking and has greatly improved the relevance of this contribution.

Many thanks, Billy J Andrews

Please also note the supplement to this comment:
https://se.copernicus.org/preprints/se-2019-202/se-2019-202-AC1-supplement.pdf

―――――――――――――――――――――――――――

---

## Author Comment (AC2) · 4 Jul 2020

**Ref: se-2019-202**

**Title: The role of pre-existing jointing on damage zone evolution and faulting style of thin competent layers in mechanically stratified sequences: a case study from the Limestone Coal Formation at Spireslack Surface Coal Mine (Changed to: "The growth of faults and fracture networks in a mechanically evolving, mechanically stratified rock mass: A case study from Spireslack Surface Coal Mine, Scotland" in the revised MS).**

**Journal: Solid Earth**

Our paper has benefitted from two reviewers, Dave Sanderson (R1) and Bailey Lathrop (R2). In our response, we refer to the manuscript that was submitted for review as the 'submitted manuscript', which has been edited following comments from the reviewers to become the 'revised manuscript'. The following tables documents each of the comments received from each reviewer and our response.

Additionally, in response to the comments by both reviewers we have restructured the manuscript. The following structural changes have been made:

- Methods section 3.2: Analysis of fault and fracture networks: The subsection of this section have been combined, with the section now titled: "Lineament mapping and network analysis".
- A subsection to section 4.1: General fracture observations has been added to clearly denote interpretation from observation (4.1.1 Order of fractures within the Murkirk 6' coal).
- Sections '4.2.2 Interaction between faults and fractures within the McDonald Limestone' and '4.2.3 Large offset faults' have been switched around in the revised manuscript to improve the flow of the paper. Additionally, section 4.2.3 of the submitted MS has been renamed 'Faults that juxtapose multiple lithologies (non self-juxtaposed)'
- A section has been added between the results and the discussion to summarise the structural evolution (Section 5: Structural Evolution at Spireslack SCM), and section 5.3 of the submitted manuscript (Effect on flow pathways) has been cut down and included as part of the discussion section 6.1.
- The discussion sections have been reworked to now cover "6.1 The effect pre-existing joints and coal cleats on subsequent deformation and network connectivity" and "6.2 The role of lithology on faulting style: self-juxtaposed vs non self-juxtaposed faulting".

We thank both reviewers for their suggestions and feel that the new structure has resulted in a greatly improved manuscript. For completeness, the reviewers' full original comments are detailed at the end of this document.

| # | Line no. (sub. MS) | Comment | Response |
|---|---|---|---|

**R1: Dave Sanderson**

Major comments

| # | Line no. (sub. MS) | Comment | Response |
|---|---|---|---|
| 1 | - | **Tidy up the text by removal of inconsistencies and more precise definition of many of the terms used.** *I have added lots of specific comments and edits on the pdf. Most importantly many terms are used without clear definition, and their usage seems to vary from one section to another.* | Thankyou for providing an in depth review of the nomenclature used in this manuscript. We agree that there are sections where the message was lost due to a lack of clarity and have taken on board many of your suggestions, along with those suggested by R2. |
|   |   | *The use of 'self-juxtaposed fault' is particularly unclear. I think we need a clear definition of this term - the usage only becomes obvious as one reads the paper. In effect "self-juxtaposed" is really being used in place of "small". "Self-juxtaposed" describes the relationship between the wall-rock stratigraphy across a fault, and is thus a "topological" term describing the wall rocks NOT the fault, i.e. a small fault can produce a self-juxtaposition of the wall rock stratigraphy.* | We agree that the definition of self-juxtaposed fault needs to be included in the main text (Line 73 to 76 of the revised MS); however, we feel that the term best describes the features we observed. A 'small' fault does not adequately describe the features; For example, a 'small' fault behaves very differently when cutting a massive sandstone unit (e.g. channel set in the Spireslack Sandstone) and acting in a 'self-juxtaposed manner' compared to the same fault cutting a heterolithic unit in the Limestone Coal Formation. As such it is the lithology and stratigraphic architecture, and not fault throw per say, that is controlling many of the observed structures. The phrase 'self-juxtaposed' is widely used, particularly in fault sealing studies (e.g. . Knai and Knipe, 1998; Gibson and Bentham, 2003; Yeilding et al., 2011; Pei et al., 2015), and we believe it is a useful concept when describing faults in mechanically layered sequences. With this in mind, and to avoid confusion, we have relabelled section 4.2.3 (now 4.2.2) from 'large faults' to 'Faults that juxtapose multiple lithologies'. |
| 2 | - | **A better presentation of the maps and data.** *Although this aspect of the paper is generally good, the diagrams are rather detailed and "formal" in their approach. There are many opportunities to provide a more visual presentation of the relationships between the data. For example, the site could be more clearly described by combining the maps in Figs 1 and 3 as follows.* | *Please note Figure 3 is figure 4 in the revised MS* While we understand the advantage of being able to see the mapped faults from the BGS 1:50,000 map on Figure 4 (previously 3), we believe that to represent the map in this way makes it impossible to compare structures visible in the high wall, and to easily cross reference the map to the fault data presented in the stereographic projections (Figure 4c) and box and whisker plots (Figure 4d). Additionally, the BGS map has been shown to have several discrepancies to the sub-surface |

**Commented [ZS1]:** To make the table less long, I would merge these into one

**Commented [BA2]:** Knai, T.A. and Knipe, R.J., 1998. The impact of faults on fluid flow in the Heidrun Field. *Geological Society, London, Special Publications*, *147*(1), pp.269-282.

Gibson, R.G. and Bentham, P.A., 2003. Use of fault-seal analysis in understanding petroleum migration in a complexly faulted anticlinal trap, Columbus Basin, offshore Trinidad. *AAPG bulletin*, *87*(3), pp.465-478.

Yielding, G., Lykakis, N. and Underhill, J.R., 2011. The role of stratigraphic juxtaposition for seal integrity in proven CO2 fault-bound traps of the Southern North Sea. *Petroleum Geoscience*, *17*(2), pp.193-203.

Pei, Y., Paton, D.A., Knipe, R.J. and Wu, K., 2015. A review of fault sealing behaviour and its evaluation in siliciclastic rocks. *Earth-Science Reviews*, *150*, pp.121-138.

| | | | |
|---|---|---|---|
| | | *This allows comparison of the observed detail with the surrounding faults mapped by BGS.* | structure that was exposed during open cast operations. We have therefore decided to not combine figures 1 and 3 (4) as suggested. |
| 3 | - | **The sequence of development of the structures.** *This is presented at four main parts of the paper: (a) a discussion of the mineralisation (Section 4.1, especially lines 194-200); (b) discussion of faults (mainly section 4.2.1); (c) interactions between faults and fractures (section 4.2.2); (d) large offset faults (Section 4.2.3). The material in these sections contains a lot of detailed information and it is difficult to relate much of this to the evaluation of the fracture sequence. In part this is caused by a somewhat inconsistent terminology, with the definition and criteria for recognition of the different fracture types being unclear. I would recommend having a separate section that clearly discussed the fracture history.* | We agree that the development of structures requires clarity and a separate section describing the deformation history would be helpful. To aid this we have done two things: firstly, Section 4.2.3 (L367-431) 'Interaction between faults and fractures within the McDonald Limestone' has been added that uses focused sections to pull apart the fracture evolution. Secondly a separate section (Section 5: Structural Evolution of Spireslack SCM (L432-444) has been added to summarise and the order of structures across the site and to suggest how this may fit within a regional context.

As part of this Figure 8 (L369) has been reworked, with the fracture statistics removed and referenced in the Supplementary information and Table 3, and replaced by focused sub-sections that highlight the age relationships (Section 4.2.3).

As outlined in our introduction paragraph, the information has been made clearer in a summary of the evolution (Section 4.1.1 for coal features, a paragraph in 4.1 and section 4.2.3 for the McDonald Limestone, and a summary of all features in Section 5).

The terminology used for the network analysis has been highlighted in the methods (lines 153 to 159 of the revised MS). Additionally, reference has been made to the difference between joints, faulted joints, and shear fractures on lines 196 to 199 of the revised MS.

We have added a discussion section 'Structural evolution of the main void at Spireslack SCM' (Section 5) that summarises the age relationships discussed in the previous sections, groups them together within a sensible structural evolution, and makes a first attempt at fitting the complex structures into the broader evolution of the Midland Valley of Scotland. Further work at the site is required before it will be possible to further this aspect of the discussion and a number of recommendations for further work have been made (see 434 to 438 of the revised MS). |

| | | | Because we have had to add this section to the discussion, the section 'Implications for growth of strike slip faults' has been removed, with aspects of this section added to section 6.2. Additionally, the detailed field descriptions of the faults that cut multiple lithologies have been moved to the supplementary information (S3). |
|---|---|---|---|
| 3a | | **Section 4.1**

*In Section 4.1 the authors recognise 4 types of fracture: coal cleats, en-echelon arrays, mineralised shear fractures and barren shear fractures. These are not clearly defined and their illustration in Fig. 2 introduces a number of different terms (joint, vein, etc.).* | The definition of cleats has been improved. Due to shear fractures being previously discussed (see 198-199 of the revised MS), and en-échelon veins a commonly used term we have not elaborated these features further. The terminology introduced in section 4.1 has been applied to Figure 3, with the ankerite vein renamed mineralised shear fracture. |
| | | **Section 4.2.2**

Point 1: Reactivation of structures
*In Section 4.2.2 there is also a lot of discussion of "lineations", which appear to be mainly of low pitch, suggesting strike-slip. The authors also mention "reactivation" of the faults. I did not get a clear picture of how they view the faults. To me these could well be early normal faults (well described from elsewhere in central Scotland since the work of EM Anderson) that were later reactivated by strike-slip (again there are good example of strike-slip faulting elsewhere in the region). Is that the view of the authors? This should be discussed more clearly.* | **Please note section 4.2.2 is now 4.2.3 in the revised MS**
Point 1: Reactivation of structures

We have attempted to clarify this section and have provided a summary of the fracture evolution for sub-sections of all three sample areas (See Figure 8). Overall, we observed 8 distinct 'age sets' that are punctuated by two phases of faulting that causes 'faulted-joints' to form. This information was previously hidden in the detail and/or missing from the submitted MS.

We do not agree that the faults studied here represent reactivated normal faults due to the complete lack of extensional lineations across much of the site, and where there is evidence of extension the orientation of the fault strands fits a T-shear for either the sinistral or dextral phase of faulting. Additionally, many self-juxtaposed faults display their full profile from tip to tip and show well developed low angle slickenfibers. In response to this comment we have included a note to this effect in the structural evolution table (Table 4). |
| | | *Point 2: Terminology*
*In section 4.2.2, the interaction of faults and fractures is discussed, mainly in terms of the 'joints'. A number of key questions largely remain unanswered (or if they are they are buried in the text and not* | *Part 2: Terminology*
This section has received considerable attention in the revised MS, and has been moved to after non-juxtaposed faults to improve the flow of the MS.
The key added sections are:
1. A fuller description of the fracture history observed away from map-, or |

*obvious to me). How are these 'joints' distinguished from 'barren shear fractures'? Do shear fractures and faults clearly displace the joints. Are there cross-cutting relationships between veins and joints? If so what are the relative ages?*

photogrammetry-scale faults (Section 4.1, lines 199 to 207 and Figure 3a). This provides a description of what the pre-faulting joint set looked like, and is expanded upon in section 4.2.3: 'Interaction between faults and fractures within the McDonald Limestone'. Although some sub-map scale shear fractures are visible in Figure 3b, these are mapped as faults in section 4.2.3. The classification of 'faults' and 'joints' has been highlighted in the text, with the key difference being whether a fracture is barren, and therefore potentially part of a connected network, or mineralised and therefore act as a sealing fracture. The age relationships are also summarised in the added discussion section (Table 4).

*Point 3: the use of abutting relationships*

Point 3: the use of abutting relationships / age relationships

*The authors use abutting and cross-cutting relationships to suggest relative ages of the different fractures. This generally works well for joints (especially where cross-cutting is rare) and veins, where cross-cutting can be used to determine relative age). The problem is that faults have displacement, therefore abutting does not work so simply and needs to take account of this displacement. The fact that many of the fractures are mineralised, should allow greater use of cross-cutting.*

*The abutting relationships in Fig 5 appear to contain many contradictions (although I agree that the faults look late). The problem is that abutting and cross-cutting work well for joints (especially where cross-cutting is rare) and veins, where cross-cutting can be used to determine relative age. The problem is that faults have displacement, therefore abutting does not work so well.*

We agree that the use of purely abutting relationships was not appropriate for this site, and have therefore reassessed the age relationships, with cross-cutting and abutting relationships for pre-existing features, and a visual assessment in consultation with field notes used for the identification of fault-related shear-fractures, faults, and faulted joints. Where contradictions occur this could be because:
a) The initial stage of jointing is somewhat complicated, and while large trace-length ~NE joints formed first, likely associated with the formation of the Muirkirk syncline, the later stages of joint formation caused several off-trend fractures to form and prior to faulting local rotation of bedding was likely. This will cause a highly heterogenous stress field and cause 'out of sequence' fractures to form. Additionally faulted joints will cause additional fractures to form (as outlined by the increase in fracture intensity between the three panels of Figure 8), and therefore locally joints may have been incorrectly assigned to a phase
b) Due to the complex pre-existing joint network, and similar fault orientations between phase 1 and 2 of faulting, faulted joints will be both common and locally difficult to assign to a phase due to Riedel shears forming.
c) Where joint-intensity is high strain could be distributed throughout many structures, with throw on any individual structure being below

that of the resolution of the drone map (c. 2.5 cm).

*Point 3b: surprising age relationships/contradictions*
*The analysis contains a number of rather surprising conclusions: 1) the "joints" predate the "faults", yet the faults are mineralized! 2) How do the faults and joints fit into the sequence of mineralization (lines 194-200)? 3) One set of "faults develops sub-parallel to the c.N110E joint set - can these be described as "faulted joints" (in sense of Zhao and Johnson (1992, JSG 14, 225-36). These are important outputs for this paper and need to be clarified and highlighted, and discussed later in the paper.*

Point 3b

1. We believe this an interesting observation can be explained through a combinations of a) the hydrological conditions at the time of flow favouring vertical flow, as evidenced by mineralisation being able to be traced between underlying layers, and b) the paleo-stress state favouring challelised flow along favoriably orientated structures, likely due to a high stress ratio (k <3) (Baghbanan and Jing, 2008) [Lines 489 to 492 of the revised MS]. It can also not be ruled out that micro-cataclasite and/or eroded out cementation was present prior to late stage uplift along the NE trending joint sets.
2. We have made reference to this in section 6.1. However, the mineralisation history of the site is still being worked out. It appears that the calcite mineralisation is associated with the stage 1 of faulting, with pyrite rich fluids common in stage 2, however, further fieldwork and geochemical analysis is required to confirm this.
3. The concept of faulted joints has been made clear throughout the paper.

*Point 4: Summary of joint/fault relationships.*
*This summary of joint/fault relationships shown in Fig 5 (lines 276-288 and Table 2) is interesting. The table has a lot of very long sentences and is difficult to read. The single sentences are really a collection of phrases that could easily be separated (and arranged more logically). There also appears to be a lot of missing words.*

*Point 4: Summary of joint/fault relationships.*
Table 2, Now Table 3 has been redrafted so as to summarise the fracture statistics of the 'joint' and fault networks outlined in Figure 8. Much of the text has been amended in relation to the reassessment of the fracture network, and the text that remains only provides a brief summary to aid the reader in understanding the fracture map.
The relative relationships have also been added as annotations to Figure 3a, an area of low fault intensity, to show how pre-existing joints interact with mineralised shear fractures.

**Section 4.3**
*The section on larger faults (4.2.3) contains a lot of detailed observation, but I would like to have seen some synthesis of these details. The discussion is not helped by the repeated use of 'offset' instead of 'separation'. Given that 'offset' can mean either a displacement*

***Please note this is section 4.2 of the revised MS***
To address this comment, along with making space in the manuscript for an expanded discussion about the evolution of fractures we have summarised the field observations into a Table 3, and condensed this section to summarise the key take home messages (L333-356).

| | | | |
|---|---|---|---|
| | | *component or a separation, it would be best to abandon the term altogether and talk about separation (where only relationships of layers is known) and displacements (where kinematics is supported by striations and/or piercing points).* | |
| 4 | - | **Fault and fracture topology**
*<point 1>*
*The topology of the different fracture types contains some clear errors. Since the procedures are not explained clearly enough, I am left to speculate as to the causes. I think the analysis using NetworkGT was carried out as follows: 1. The entire network was digitized. 2. The data were divided into sets based on orientation and fracture type (faults and joints) 3. The nodes and branches for entire network were then calculated (using the tool in NetworkGT) 4.* | *<point 1>*
We agree that the methods section here is not explicit enough to convey how we undertook the analysis. The most important point was that the faults and fractures were mapped separately, in conjunction with field observations, and not, as speculated, the whole network digitised then split into fault and joint sets later. This was done 1) to reduce the subjective bias, which is increased when lineament mapping is undertaken only from field photographs (or drone imagery) (See Andrews et al., 2019), and 2) ensuring that features with no stratigraphic offset were not mapped as faults accidently based purely on orientation alone.

To increase the clarity of our methods we have restructured the methods section 3.2 and renamed it 'Lineament mapping and network analysis'. Section '3.2.1: Mapping procedure' has been removed from the resubmitted MS. We have moved the text that introduces topology to the 1st paragraph of the section (Lines 133 to 140 of the resubmitted MS), and added an explicit stage 'stage 1: Lineament mapping' (Lines 148 to 159 of the resubmitted MS) that clearly outlines how the datasets where digitised, and then merged to create a 'combined network' shapefile.
We then treat all three networks as separately using steps 2 to 4 and have clearly stated this on Line 157 to 159 of the resubmitted MS. We believe this is a logical way to assess the network, and reduces the level of subjective bias in our dataset caused by the miss-identification of faults based purely on orientation. Additionally, using this method we can assess the paleo-connectivity recorded in the mineralised fault strands that cut the McDonald Limestone pavement, and assess how these will effect modern day network connectivity. |
| | | *<Point 2>* | *<Point 2>* |

*This would then give the correct results for the entire network. This looks to plot correctly on Fig. 8, as the diamond symbol, although these are erroneously assigned to the "fault strands" in the key. The correct assignment would be: squares – faults; circles - open fractures and diamonds – all network, as indicated on the node triangle..*

*The resulting values for Pc in Table 3 also appear to be correct (i.e. 0.96 ☐ Pc ☐ 1).*

*To get the corresponding data for the faults and joints, the above steps should have been repeated for each fracture type. This is clearly not what was done, since the joints would have given very similar results to the entire network, i.e. high Pc and Y-node dominated. There are certainly not as high a proportion of I and X nodes as plotted in Fig 8. The results for the branch plots may be more robust.*

*I think that the nodes for the entire network were somehow distributed into values for the joints and faults, probably by removal of the fault-related nodes and assigning those that remain to the joints. This does not work because the majority of the nodes are produced by intersection with joints. I think there may be similar problems with splitting the fracture types into sets, as individual sets of sub-parallel fractures contain*

Figure 8 (Now Figure 9) was indeed incorrectly labelled in the Key, something that was not picked up in pre-submission edits. The correct symbology should indeed be combined network = diamond, 'open fractures' = circles, and 'combined network = diamond. This has been corrected in the revised MS.

The values in Table 3 represent the values extracted from the three sample areas, the incorrect values were apparent in the figure. I hope the correct notation in Figure 9, along with the description of our methodology aids in clarifying this section. Table 3 of the submitted MS has been removed, with readers directed to the supplementary information for fracture statistics of the combined network, and Table 3 for the key statistics of the 'joint' and fault network.

To get the data for the faults and joints, the sample areas were analysed using the originally digitised networks. In the joint network (a) the high proportion of i-nodes arises from the abutting relationships between joints and faults, that are themselves mineralised and act as an i-nodes. In the combined network, many of these connections are represented as y-nodes and as such the ratio of y-nodes decrease, and i-nodes increase in the joint network compared to the combined network (c). This was also compounded by the joint data for Fig. 8a being incorrectly plotted in Figure 9 (the data-point was mirrored), which has been corrected in the revised MS.

This is evidenced by looking at the figures for Sample SA3 [Fig. 5c] (Supp2). In this sample area the fault network is dominated by I (729) and y (593) nodes, with very few x-nodes (6). Conversely, the fracture network, where fault and fracture connections are classed as i-nodes is broadly split between i- (4517) and y- nodes (4726), with a 291 x nodes. However, if the network was digitised as a whole, with faults and fractures both included (as with the combined, or 'full' network), as is often the case in remote sensing fracture studies, then the connection between joints and faults would become y-nodes. Additionally where a fault with very minor offset that is below a pixel resolution (e.g. a fault tip) cuts a joint,

| | | | |
|---|---|---|---|
| | | *few connected nodes and very few I-I branches (c.f. as plotted).* | this would be classed as an x-node. Therefore, explaining the increase in x-nodes (607), decrease in i-nodes (208) and increase in y-nodes (10266). Because the faulting intensity is higher in SA3 than SA1 and SA2, the effect is more pronounced, and in SA1 the decrease in nodes is less pronounced. The same trends are observed in the branch data. I hope that the explanation of our digitization method improved the clarity of why we undertook this approach. We therefore do not agree that Figure 8 and Table 3 are flawed. We have changed 'Full network' to 'combined network' throughout the manuscript to aid the reader. |
| | | *This means that the discussion of the implications of flow based on Fig 8 and Table 3 is flawed.* | |
| 5 | - | **Network properties and flow/permeability** *There is also another major flaw in the discussion of flow and permeability in relation to the network characteristics. The reason this is flawed is that the network topology, essentially evaluates the connectivity of the fractures, whereas the permeability of the rock mass is a step-like function dependent on the percolation threshold, and, once this is reached, primarily on the conductance of the fractures on the connected component of the network. The correct topological analysis of each fracture type would allow discussion of this, but it would need to recognise the difference between the rolls of: (a) a conductive joint network, (b) the palaeo-flow in an active fault network, and (c) the subsequent effect on mineralization of the fault network. For example, superposition of a connected network of sealing faults would counteract the conductivity of an earlier network of open fractures.* *This whole section talks about "open fractures". These are not clearly defined. How do they correspond to the 'barren shear fractures' and 'joints' discussed earlier.* | We agree that permeability is not the correct term to use in this context and have changed this to 'connectivity' throughout the manuscript. We have re-structured the discussion, to us topology to to provide information about fault growth in jointed sequences. Where we do discuss connectivity (section 5.1 'Modern day network connectivity' of the revised MS), we have made it clear we are not discussing permeability, but network connectivity. |
| 6 | - | **Abstract and conclusions** | Sections have been rewritten to more align to the evolution of the described features. |

| | | | |
|---|---|---|---|
| | | *These do a very poor job in summarising the content of the paper. There is too much "discussion" and arm waving about role of lithology, strengthening by mineralization, and effects on permeability, with not enough on the key issued of the sequence of development and interaction of the different fracture elements.* | |
| Minor comments | | | |
| 7 | 2 | Rather long title | We agree and feel that the title give a poor representation of the MS. Therefore we have changed the title to "The growth of faults and fracture networks in a mechanically evolving, mechanically stratified rock mass: A case study from Spireslack Surface Coal Mine, Scotland" (L1-3) |
| 8 | 24-26 | I agree that this subject has advanced greatly in recent years, but the role of layering was recognised much earlier, with key papers such as Anderson 1951, Donath 1961 Ranalli & Yin 1990 - see Peacock and Sanderson 1992 J. Geol. Soc. Lond. 149, 793-802. | We agree that a comment on the earlier work is relevant here, and have included a number of the references that you suggest. (L25-26) |
| 9 | 42 | Again these "Yoredale" cycles have been described for decades. | We agree that these are covered in many publications, however, feel that the summary provided in the text book Thomas (2012) provides a good overview that is sufficient for the purposes of this submission. Seminal references are included in the textbook and hence have added 'and references therein' for interested readers. (L59-60) |
| 10 | 54 | I think we need a clear definition of this term - the usage only becomes obvious as one reads the paper. In effect "self-juxtaposed" is really being used in place of "small".

"Self-juxtaposed" has an interesting implication as it describes the relationship between the wall-rock stratigraphy across a fault, and is thus a "topological" term describing the wall rocks NOT the fault, i.e. a small fault can | See point 1 above. We have added this to the final paragraph of the introduction, along with linking it to where it has been previously used and stating how we use it. (L73-76)

We believe that this is a key concept, in addition to considering the temporal evolution. We have therefore extended this in our discussion (Lines 542 to 567). |

| | | | | |
|---|---|---|---|---|
| | | produce a self-juxtaposition of the wall rock stratigraphy. | | |
| 11 | 61 | Change to 'faults' | Changed, thankyou (L84) | |
| 12 | 70 | "abandoned" refers to coal mining, but not village?? | While many coal mining villages live on, Glenbuck is one of the few that actually disappeared when sub-surface mining ceased. The corner flags used by the Liverpool FC legend Bill Shankly are now marked by plastic tubes in a peat bog. | |
| 13 | 72 | Remove 'slope along' | Removed, thankyou | |
| 14 | 96 | This not in references - needs greater clarification and attribution. 'Bing (2017)' | Attribution of the source has been included in the revised MS, Thank you for pointing this out. (L740-741) | |
| 15 | 96 | 'offset' : do you mean 'offset' or 'stratigraphic separation' or something else? | I have changed to 'stratigraphic offset' as there are cases where true offset is either impossible, or very difficult, to quantify (e.g. L130). | **Commented [ZS3]:** Is this not the same as throw? what is stratigraphic offset – try not to invent new terms! |
| 16 | 101-110 | I think the term "high wall" need some explanation, and a consistent spelling. | Have added in a little more detail about the geometry of the open cast mine including a definition of 'high wall'. The spelling of high wall has also been standardised, thankyou and R2 for pointing this out. Additionally we have added a figure to aid the reader visualise the site (Figure 2, L98-102) | |
| 17 | 110 | Not in references! Is this a published paper? | The reference is on the submitted manuscript, line 612-614; However, the DOI was incorrect on the submitted MS and has been updated in the revised MS. (it was the DOI of the discussion paper on Solid Earth and not the published MS). (L609-611) | |
| 18 | 112 | The branch/node model is a way of describing one aspect of the topology. | Amended the opening sentence of section 3.2 to address that this is one topological method. (L133-140) | |
| 19 | 115 | Sanderson and Nixon use I, Y, X, and you revert to capitals in Fig 8 - need to be consistent. | Changed to capitals in the text to be consistent with Sanderson and Nixon (L135-136) | |
| 20 | 119 | Explain or refer to later Equation 1. | Referred to Eq1 in the text. (L174) | **Commented [BA4]:** Check where this is in the submitted MS |
| 21 | 137 | 'Percentage' to 'proportion' | Changed, along with changing percentage to proportion (L136 & L172-173) | |
| 22 | 145 | use subscripts - as is conventional and in papers cited. | Amended. (L175) | |
| 23 | 158 | The fracture sets in most subsequent diagrams are NOT typically arranges in orthogonal | Thankyou for pointing this out, please see the reply to R1 Comment 3a. | |

| | | | |
|---|---|---|---|
| | | sets. Indeed in the next sentence you imply this. You then have a NNW set in the following sentence. I think "orthogonal" is a totally inappropriate term to use for what is clearly a complex fracture history, and certainly does not imply that a "cross-joint" pattern, widely reported for joints. | |
| 24 | 206 | stress is a symmetric 2-order tensor, and should not be described - any reference to 0"dextral" must involve some interpretation of the strain and/or rotation. | Removed the reference to stress, thankyou. |
| 25 | 220 | The observations in this section, based on careful mapping and evaluation of abutting relationships, is the real value of this work. It clearly indicates initial description of an orthogonal network is NOT the case. I would avoid setting up "straw men" only to demolish them. | We agree that the initial description of the fractures presented in L158 is not satisfactory and that a reworkings of this section will greatly increase the relevance of this work. We have reworked this section to remove the 'straw men' caused by 'orthogonal joints'. |
| 26 | 238 | 'lineations': Can you be more precise? Are these slickenfibres?  Why are no data for these linear features presented? | Lineation data have been added to Figure 3c, originally they were removed for clarity of fault orientation. |
| 27 | 241 | 'belong to' to 'Have' | This sentence was removed during the restricting of the revised MS. |
| 28 | 257 | I do not see a "strain ellipse in Fig. 3. Again an ellipse should not be described as sinistral or dextral, but can be interpreted as resulting from some sheared zone. | The wording has been changed to "The majority of faulting at Spireslack SCM fits into the expected fault geometries for Riedel shears under a sinistral shear sense (Figure 4c)." |
| 29 | 261 | Be careful! Did we learn nothing from Ramsay - folds imply strain NOT stress. | Removed the suggestion of stress throughout the MS. |
| 30 | 263 | 'Strain' to 'shear' | Changed, Thankyou (L278) |
| 31 | 270 | joints or veins. Remember, a fault is also a fracture! | We have made it clear that in section 4.2.3. joints refer to both joints and barren shear fractures, see R1 comment 1. |
| 32 | 289 | This table has a lot of very long sentences and is difficult to read. The single sentences are really a collection of phrases that could easily be separated (and | Table 2 has been removed from the revised MS, please see R1 Comment 3. |

| | | arranged more logically).  There also appears to be a lot of missing words. | |
|---|---|---|---|
| 34 | 317 | so some joints postdate faulting! | This has been clarified in the text, please see the reply to R1 Comment 3. |
| 35 | 339-341 | Font change! | Changed, Thankyou. |
| 36 | 365 | We seem to skip "Example 5"! | As part of the restructure of this section, detailed observations have been moved to the supplementary information (S3). We have corrected the missed example in this text. |
| 37 | 366 | Or is it cut by fault | While it appears that the fault cuts the dyke, there is no evidence of dyke material within the fault-bounded lens, and while white-trap is well developed outside of the fault zones, the same cannot be said for the coal and organic-rich mudstone within the fault-bounded material. It is therefore our interpretation that the dyke breached the fault out of the plane of observation, and could not cut through the widened fault core visible in the high wall. |
| 38 | 373 | 'offsets' to 'seperations' | Changed to separations, thankyou. |
| 39 | 388 | This section on larger faults contains a lot of detailed observation, but I would like to have seen some synthesis of these details.  The discussion is not helped by the repeated use of 'offset' instead of 'separation'. Given that 'offset' can mean either a displacement component or a separation, it would best to abandon the term all together and talk about separations (where only relationships of layers is known" and displacements (where kinematics is supported by striations and/or piercing points. | We have taken on board your suggestion to remove the use of offset, and change over to the use of either 'stratigraphic offset' or 'stratigraphic separation'. Due to the wide spread of lineation measurements within a single fault strand we believe it would be unwise to attempt to calculate displacement without a focused field campaign collecting this data. This was not possible at this site due to access restrictions during the 2018 and 2019 field seasons. |
| 40 | 396 | I guess this means the bringing together of different lithologies. Is it not just "justaposition, and does it not apply to the wall-rocks rather than the fault. | Please see the response to R1 comment #1 |
| 41 | 398-399 | Does this imply faults in incompetent units can not be "self-juxtaposed" ? | I don't believe this to be the case, and it is the thickness of the layer that controls whether something is "self-juxtaposed". Where shale contains limited strength barriers, then I would argue that larger faults would self-juxtapose these lithologies; however, self-juxtaposition in heterolithic sequences is |

Commented [BA5]: Do I need to add some text in the MS for this? I would imagine it would help the reader, however, it was taken out of an earlier draft.

Commented [ZS6]: And again here

| | | | limited to very low offset faults (offsets smaller than layer thickness). |
|---|---|---|---|
| 42 | 411-412 | This ignores the role of fluid pressure. What is important is the ability of fractures to form. | We agree fluid pressure is important, however, we believe there is no way to know what the influence of pore fluid pressure was in this case. Shales can be very over pressured within fluvial deltaic environments. We therefore have decided not to include this in the discussion. |
| 43 | 419 | Meaning unclear. | This part of the discussion was found to distract from the key points of the manuscript and has been removed from the revised MS. |
| 44 | 420 | 'in, where' to 'of' | This part of the discussion was found to distract from the key points of the manuscript and has been removed from the revised MS. |
| 45 | 420-421 | Remove 'coals in the form of' and 'often' | |
| 46 | 452 | 'likely' | |
| 47 | 455 | see also work of Jones and Tanner (1995 - JSG 17, 793-802). | The work of Jones and Tanner provides some useful context for the earlier development of the basin, and it is likely that some of the Devonian deformation carried through to the earliest Carboniferous, however, evidence at the site is that the observed deformation occurred post-diagenesis and therefore more likely related to the later Carboniferous deformation outlined by Leeder etc. As such we have added the reference to the discussion regarding shear strain localisation along the pre-existing weaknesses (L458-459). |
| 48 | 458-549 | see also Jones and Tanner 1995 | See above |
| 49 | 476 | 'Bense et al., 2013' Not in references. | This reference has been removed from the revised MS |
| 50 | 515 | How does this figure support this statement? | As part of the redrafting, this sentence was removed. Please see the introductory paragraph to our reply. |

Commented [BA7]: Zoe could you have a quick look at this, I am a little unsure where Dave Sanderson wants this reference included. It is a great paper I was just not sure what point of it he wanted included.

| \multicolumn{4}{|c|}{R2: Bailey Lathrop} | | | |
|---|---|---|---|
| \multicolumn{4}{|l|}{Major comments} | | | |
| # | Line no. (sub. MS) | Comment | Response |
| 1 | - | The writing style lacks clarity and is often difficult to read and understand. | Several sections have been edited to improve the clarity of the writing, with terminology standardised throughout the MS. |
| 2 | - | The results section should be reorganised for clarity, and interpretations and results should be split | Sections of the results have been restructured (S4), and we have added a synopsis of results for faults that juxtapose multiple lithologies (Table 2). Additionally we have added a section that pulls together the order of events (Section 5). For more detail please see our introductory paragraph and reply to R1 comment 1 & 3. |
| 3 | - | The paper needs a though proofread | The MS has undergone a proof read, and several cases of inconsistent spelling has been rectified. |
| 4 | - | The figures are too busy, and could be simplified. | Some of the figures have been simplified as outlined below: Fig. 1: The boxes have been rearranged to enlarge the geological map surrounding the main void and to enable text size to be increased. Additionally, the names of the coal fields have been removed. Fig 2: Added to provide the reader with a visual of the site and to clarify the void terminology (e.g. High wall, dip-slope etc.) Fig. 3, 5, & 6: Boxes around annotations have been removed to clarify the field photographs and highlighted features. Fig 5: This figure has been redrafted to better describe the fracture evolution across the limestone pavement, please see the reply to R1 Comment #3. |
| 5 | - | Relevant figures need to be referenced in the text more often | Increased the cross-reference to figures throughout where it was deemed appropriate to do so. |
| 6 | - | The abstract and conclusions should contain more specific results | Both the abstract and conclusions have been redrafted in the revised MS. |
| \multicolumn{4}{|l|}{Minor comments} | | | |
| \multicolumn{4}{|l|}{Thankyou for suggesting many textual changed to the MS, we have taken many of these on board in the revised MS.} | | | |
| 7 | 1-3 | Maybe combine the two sentences in the title. | The title has been changed to better represent the focus of the MS |
| 8 | 34 | Could this be a second paragraph specifically about pre-existing | We have split the paragraphs to improve clarity as suggested (L39-49). Additionally, we have |

| | | weaknesses? It's an important and separate point from the sentences on mechanical stratigraphy. Alternatively, add something about pre-existing weaknesses to the opening sentence of the paragraph. | added a paragraph to introduce the effect of cementation on the evolution of fracture networks (L50-57) |
|---|---|---|---|
| 9 | 36 | Maybe explain what you mean by stress ratio | Stress ratio is the ratio between the minimum and maximum stress and represents a well known concept in fracture mechanics, we therefore don't believe further elaboration is required for the readership of Solid Earth. |
| 10 | 41-42 | This is a very british term, and I don't think many people will know it. Can you explain what it is briefly here, like this:

Fluvial-deltaic sequences are characterised by cyclical sequences of limestone, sandstone, siltstone, seat-earth (i.e. XXXXXX), shale, and coal | Added '(i.e. paleosols that are often found beneath coal seams)' (L59-60) |
| 11 | 45 | Doesn't read quite right. Maybe say: Cleats form coals as diagenesis takes place, which creates pre-existing weaknesses that may affect the location, orientation, and length of faults. | The sentence has been amended and a reference added with respect to the formation of cleats. The references that were previously in the sentence have been moved up to the 'pre-existing' paragraph so as to not suggest the work is on the role of cleats on faulting. (L61-64) |
| 12 | 54 | Does everyone know what a self-juxtaposed fault means? | Please see reply to R1 Comment 1. |
| 13 | 61 | 'tectonic lineaments' (comment unclear on PDF, check if comes up on Adobe). | Changed to 'faults' (L84) |
| 14 | 87 | Font in this figure is a little too small and difficult to read | Please see our reply to R2 comment #4. |
| 15 | 103, 106 | how do you spell this? | The spelling of 'high wall' is now consistent, thank you. |
| 16 | 138 | Perhaps add info on where this comes from and what it means | Added reference in the main text. |
| 17 | 145 | What does this mean? What are the variables? | The definition of the variable has been added along with the units. (L182) |
| 18 | 148 | Make all of the text consistent. In 4.1a, Late should be capitalized because for consistency. Same with Brecciated in 4.1e and En-echelon in 4.1f . | Changed, thank you (Fig 3) |
| 19 | 158 | Are they orthogonal?... | This has been removed from the resubmitted MS, please see the reply for R1 Comment 22. |

| 20 | 189 | How can you tell that they weren't connected to a source of mineral fluids? Make that clear. | Changed, thank you (L480-492) |
|---|---|---|---|
| 21 | F5 | Some of this text is too small to read | This figure has been redrafted, please see R1 comment 3. |
| 22 | 270 | Label the figure as Fig. 5a-c so that you can reference it more directly in the text | This has been incorporated into the revised MS, thankyou. Note: Figure 5 is Figure 8 of the revised MS. |
| 23 | 281 | Say specifically 5a | This section has been reworked in the revised MS and now clearly splits the observations from each sample area (please see lines 369 to 428 of the revised MS). |
| 24 | 283 | Fig 5b | |
| 25 | | Table 2 | Table 2 has been reworked, please see R1 Comment 3a |
| 26 | 400-401 | See my notes on this | Due to the focus of this paper, we have not explored the fault growth models in detail. To undertake this work we would need to have collected more D-L data from the faults, which was not possible within the constraints of the available field time. We would be happy to discuss the implications of this work, and potentially further work into this area. We have alluded to the distribution of fault rock matches the model of Childs et al 2009 on lines 550-553, and that normal models appear to match on lines 559-562, however, more data is required to expand this discussion. |
| 27 | 402-403 | Interesting! | Thank you, we think it is an important point in the deformation of shale-rich mechanically stratified successions. |
| 28 | 453 | Formed | This section has been moved into the table in section 5 (Table 4). The tenses have also been corrected. |
| 29 | 454-455 | Combine with previous sentence | |
| 30 | 484 | Can you say what you mean by 'large'? Like (>10 m) or whatever. | This part of the discussion has been reworked in the revised MS. |
| 31 | 494-496 | Take a look at Rotevatn et al., 2019 paper on fault growth models. What you are describing is more of a isolated fault growth model, as I mentioned in another comment. We rarely see growth like this (granted in normal faults), so this is interesting! | Please see the reply to R2 comment #68 |
| 32 | 514-515 | Can you explain Fig 8 a little more? | This figure is now Figure 9: A further description of the figure is now provided in the figure caption (L511-515). |
| 33 | 527 | units? | This section has been removed from the revised MS, units have been included throughout the discussion of connectivity (L493-509). |
| 34 | 531 | = to be like the previous one | Changed thank you (L502) |
| 35 | 561, 565 | Dip slip, dip-slip | This section has been removed from the revised MS. |

| 36 | 568-573 | This repeats a bit. Could be re-worded. | This section of the discussion has been removed from the revised MS. |
|---|---|---|---|
| 37 | 577 | Reword | |
| 38 | 581 | The effect of what? Be more clear. | |
| 39 | 583 | ? | |
| 40 | 583 | Don't use the word significant, as it's a statistics term | |
| 41 | 587-588 | Are you trying to say growth faults or growth OF faults. Make that clear. | The conclusions have been reworked to better focus on the key points of the MS. |
| 42 | 603 | I think this would be a more powerful last sentence without the /s | The final sentence of the MS has been changed to 'Therefore, it is crucial to appreciate the relative timing of deformation events, concurrent or subsequent cementation and the degree of lithological juxtaposition when considering the mechanical and hydraulic properties of a mechanically stratified succession.' to better represent the findings presented in the manuscript [L596-598]. |

---

## Referee Report (RR1)

**Review of paper submitted to Solid Earth**

**Title**: The growth of faults and fracture networks in a mechanically evolving, mechanically stratified rock mass: A case study from 3 Spireslack Surface Coal Mine, Scotland. **Authors:** Billy J. Andrews, Zoe K. Shipton, Richard Lord, Lucy McKay

This paper has been substantially revised from the earlier version and provides a detailed description of a very interesting exposure in central Scotland that should be published. The clarity of the maps and diagrams has been improved. The text is a bit more concise and better structured, and there is much better definition of the terms used. The discussion is more focused on the evolution and connectivity of the fractures, with removal of previous discussion of "permeability"

I still find a number of aspects of this paper difficult to follow, and have made many comments on the annotated text. My main concerns are:

- The term 'self-juxtaposed fault' as "self-juxtaposed" describes the relationship between the wall-rock stratigraphy across a fault. This should be "self juxtaposing" to be logically and grammatically correct. A fault that juxtaposes different lithologies is later referred to as a "non-self-juxtaposed fault" - does this really help communication of the simple idea that faults may juxtapose the same or different lithologies?
- 2) The use of "Riedel shear" (lines 277-2870 seems both un-necessary and at variance with the original description of fractures developed above basement shears.
- 3) The authors use abutting, cross-cutting and off-set relationships to suggest relative ages of the different fractures and faults. This results in 8-10 "phases" of fracturing, that they relate to 4 (or 5) stages of development (Table 4). Aspects of this analysis are clearly documented with field photos and summarized in Fig 8. Many of the off-set relationships used to infer faults post-dating joints vary along the faults, with some large offsets close to fault tips. I think some of these features could be interpreted as "trailing".
- 4) The authors defend the interpretation of joints (unmineralized) pre-dating faults and veins (mineralized), but I still find this odd.
- 5) There is still inconsistency and lack of clarity in the topological analysis; the procedures are still not explained in sufficient detail. For example: (a) Nodal % in Table 3 do not sum to 100%; (b) I cannot calculate the given Pc from numbers of nodes using equation 1 (excluding E nodes); (c) I still do not see how these node counts were done. It looks as if a lot of the fault I-nodes are in fact Y-nodes with termination of the fault at a joint (and *vice versa*).

David Sanderson September 2020

[revised manuscript text omitted]

---

## Author Response (AR2)

**Ref: se-2019-202**

**2nd round of review**

**Title: The role of pre-existing jointing on damage zone evolution and faulting style of thin competent layers in mechanically stratified sequences: a case study from the Limestone Coal Formation at Spireslack Surface Coal Mine (Changed to: "The growth of faults and fracture networks in a mechanically evolving, mechanically stratified rock mass: A case study from Spireslack Surface Coal Mine, Scotland" in the revised MS).**

**Journal: Solid Earth**

During the 2nd round of peer-review our manuscript received one reviewer report from David Sanderson that pointed out a number of minor edits that were missed during the previous rounds of editing. Please find below our point by point response and we would like to thank David Sanderson for his detailed and thoughtful reviews that have significantly improved the manuscript. We hope the completed edits mean that that the paper is ready for publication,

Many thanks,

Billy J Andrews (on behalf of the authorship team).

| R1: Dave Sanderson | | | |
|---|---|---|---|
| Major comments | | | |
| # | Line no. (sub. MS) | Comment | Response |
| 1 | - | The term 'self-juxtaposed fault' as "self-juxtaposed" describes the relationship between the wall-rock stratigraphy across a fault. This should be "self juxtaposing" to be logically and grammatically correct. A fault that juxtaposes different lithologies is later referred to as a "non-self-juxtaposed fault" - does this really help communication of the simple idea that faults may juxtapose the same or different lithologies? | We appreciate that grammatically "self-juxtaposed fault" is incorrect and instead of the fault itself being juxtaposed it is the lithologies/facies on either side. As such, we have taken your suggestion to change the term to "self-juxtaposing". We however, do feel that the concept of self-juxtaposition is a very useful concept within mechanically layered sequences and have therefore not removed the term altogether from the MS. |
| 2 | - | The use of "Riedel shear" (lines 277-287) seems both un-necessary and at variance with the original description of fractures developed above basement shears. | The mention of Riedel shear has been removed from this section of the text as suggested. |
| 3 | - | The authors use abutting, cross-cutting and off-set relationships to suggest relative ages of the different fractures and faults. This results in 8-10 "phases" of fracturing, that they relate to 4 (or 5) stages of development (Table 4). Aspects of this analysis are clearly documented with field photos and summarized in Fig 8. Many of the off-set relationships used to infer faults post-dating joints vary along the faults, with some large offsets close to fault tips. I think some of these features could be interpreted as "trailing". | While the presence of trailing segments would enable the simplification of the deformational history, and a reduction in the number of required phases, no direct evidence was observed during fieldwork. This however does not discount there being subtle evidence (e.g. lineation's along pre-existing joints) that was missed and therefore we have added the following line to the to the revised MS:

"This may be due to the development of 'trailing segments' (i.e. sections of a previous structure reactivated during subsequent deformation (c.f. Nixon et al., 2014)), however, no direct field evidence was observed as part of this study (e.g. mineralisation and/or evidence of shear)." [Line 431-434].

And added the following to the figure caption of Fig. 8:

"Please note that while it is possible some joint's and/or faults acted as trailing segments (cf. Nixon et al., 2014) no direct field evidence was observed."

Despite the effect trailing segments would have on the deformational history (i.e. a reduction in the number of required stages), |

| | | | |
|---|---|---|---|
| | | | we stand by our interpretation of 4 primary deformation phases, that are also preserved in other aspects of the site (e.g. the coal and faults) of 1) pre-existing joints, 2) joints and faulted joints related to early sinistral shear, 3) joints and faults related to dextral shear and 4) minor joining that post-dates the two primary deformation phases. |
| 4 | | The authors defend the interpretation of joints (unmineralized) pre-dating faults and veins (mineralized), but I still find this odd. | While we appreciate the reviewers view, which is shared by others in the fracture community, it is the opinion of the authors that mineralisation will occur only when the chemical and hydrogeological conditions are favourable. A fluid will only flow along the 'path of least resistance' and hence can bi-pass significant parts of a network (as discussed on lines 494 to 506). An additional point is that mineralisation will only occur when then chemical conditions are correct to do so (e.g. through a rapid drop in pressure). |
| 5 | | There is still inconsistency and lack of clarity in the topological analysis; the procedures are still not explained in sufficient detail. For example: (a) Nodal % in Table 3 do not sum to 100%; (b) I cannot calculate the given Pc from numbers of nodes using equation 1 (excluding E nodes); (c) I still do not see how these node counts were done. It looks as if a lot of the fault I-nodes are in fact Y-nodes with termination of the fault at a joint (and *vice versa*). | (a) Thankyou for pointing this out! It seems that during the redrafting of T3 that the number of nodes did not carry across correctly from the excel file. This has been amended and for clarity the # of nodes has been included, with nodal % included in a bracket. The rest of the table has been checked against the extracted data and is correct. This will also explain your point (b), with Eq 1 now returning the Pc as detailed in the table.
(c) We feel that a further explanation of node counting is indeed required and you are correct in your point that fault i-nodes and joint i-nodes can represent y-nodes in the combined network. The rational and explanation of our methods are as follows:

"The digitisation and analysis of the fault network separate from the 'joint' dataset meant that where faults terminated against pre-existing joints (i.e. a y-node in the combined network), this was classified as an isolated node. This was done to provide the network properties (i.e. connectivity, trace length and fracture intensity) of the 'active' fault network where evidence of shear and mineralisation is present. Because the mineralised fault network will be sealing to flow, and therefore not hydrologically connected to the joint network, it is not |

| | | | |
|---|---|---|---|
| | | | appropriate to classify joint-fault abutting relationships as connected nodes. Therefore, where a joint terminates against a pre-existing fault in the 'joint' dataset this was also classified as an i-node. The combined network represents the fault-fracture network that is typically digitised and analysed for topological analysis." [Lines 463 to 473 of the revised MS].

We have also added the following to the figure caption of Table 3:
"Please note, because the fault network is superimposed onto the joint network, i-nodes (i.e. where a fault terminates) can represent a y-node in the combined network. Similarly, where a joint terminated against a fault, due to the sealing properties of the fault, it is no longer appropriate to classify this as a connected branch and as such is classified as an i-node in the 'joint network'." |
| Minor comments | | | |
| 7 | 58-76 | It would be good to link these two paragraphs, as it only becomes clear why the first is included in the introduction after one reads the second. Detail repeated later in the paper need not be given here.
*Otherwise this is a very clear introduction to the paper.* | Thankyou for your suggestion, we have merged these two paragraphs, moved the definition of self-juxtaposing faults to the methods section [line 121 to 125 of the revised MS] and removed some of the detail.

[For revised paragraph please see lines 58 to 69 of the revised MS] |
| 8 | 84-91 | "Midland valley" does not need to appear in every sentence of this paragraph. | Thankyou for your suggestion, this has been amended in the revised MS. |
| 9 | 182-183 | Variables are best expressed by a single letter; thus "tl" could be simply "t" as it is not the product of two varaibles "t" and "l".

"Area" is not easily mis-interpreted, but might be better a "A" for similar reasons. | To avoid confusion with t = time, we have changed the trace length variable to *L* and as suggested changed Area to A.

[please see Line 186 of the revised MS] |
| 11 | 196-207 | CRITICAL OBSERVATION OF AGE RELATIONSHIPS

IS IT CORRECT???? | The description of the age relationships provided in this paragraph is consistent with field notes, photographs and maps/sketches made during fieldwork. It is possible that 'tailing' (cf Nixon et al., 2014) occurs locally, however, no direct evidence (e.g. mineralisation stepping along barren joints and/or shear evidenced along joints) was noted in the field. |
| 12 | 305-310 | I see no reason why a fault with some constant stratigraphic separation (s) could /would not juxtapose layers with | Please see our response to major comment #1. |

| | | | |
|---|---|---|---|
| | | thicknesses t << s and self-juxtapose layers with t >> s.

As a result, I still find this use of "self-juxtaposed FAULT" odd as it is the LAYERS that are juxtaposed or not. At least "self-juxtaposing" would be a more correct adjective for "fault"

A fault that juxtaposes different lithologies is later referred to as a "non-self-juxtaposed fault" - does this really help communication of the simple idea that faults may juxtapose the same or different lithologies? No wonder the rest of the science community often thinks geology is little more that a heap of terminology and coloured maps! | |
| 13 | 363 | The mapped relationships appear to show the dyke off-set by the fault, with similar fault-parallel separation of dyke and rock layers. This needs to be discussed more carefully. There is some discussion of this in Table 4, but a lot of emphasis is being placed on not finding dolerite in the fault rocks. | We have expanded our discussion slightly in Table 4 to outline the evidence and our interpretation into how the dyke will have likely intruded. "No fragments of dyke are observed within the fault core in Fig 7a and no white trap is observed in the coal within the fault. This provides evidence that the tertiary dyke, that post-dated faulting did not intrude along the fault plane. Instead, it is likely that the dyke either injected around the tip of the fault, or broke through the fault core out of the plane of observation." [Table 4]

Additionally, we have added the evidence of white trap to the figure caption "A later Paleogene dyke, associated with the British Tertiary Igneous Provence, intrudes across the fault, however, no evidence of white trap or dyke material is observed in the fault core [See table 4 for discussion]" [Lines 366-368] |
| 14 | T3 | Why do % of node types not sum to 100? | Thankyou for pointing this, this was an error in the redrafting of T3 from the extracted excel data. This has been remedied in the revised MS. |
| 15 | T3 | These should this be 8a, b c - as indicated in caption.. | Thankyou. This has been fixed in the revised MS. |
| 16 | 429-431 | These appear to be the sample areas in Fig 8; column 1 needs updating.

I do not see how joint sets 0-4 and faults 0-4 relate to the "phases" in Fig. 8. | Thankyou, this has been updated

The sets presented in Table 3 do not relate to the phases in Figure 8. Instead "Trace length data is presented as orientation sets, that were derived following visual assessment of length weighted rose |

| | | | |
|---|---|---|---|
| | | If the values for nodes ore %, why do (I+Y+X) not sum to 100? I cannot calculate the given Pc from numbers of nodes. This should be based on equation 1 and not include E nodes.

I still do not see how these node counts were done. It looks as if a lot of the fault I-nodes are in fact Y-nodes with termination of a fault at a joint.

The caption to this table needs a lot more explanation. | diagrams, and do not relate to the age sets outlined in Figure 8.” The text in quotations has been added to the table caption to improve clarity.

Please see the response to major comment # 5.

Please see the response to major comment # 5. The following text has been added to the table caption to make it clear that this is the case. “Please note, because the fault network is superimposed onto the joint network, i-nodes (i.e. where a fault terminations) can represent a y-node in the combined network. Similarly, where a joint terminated against a fault, due to the sealing properties of the fault, this will be classified as an i-node in the ‘joint network’.” |
| 17 | 435 | ‘void’ ? | Removed from the revised MS. |
| 18 | 451-452 | There is a long history of doing this and it seems odd to cite these two papers. Both these papers are careful to abutting relationships to deduce sequences of fracture development and NOT to assume these relate to specific tectonic events. In the case of faults (and possibly some joints) more than one orientation set may be produced in the same deformation event. This, together with local stress changes during fracturing (e.g. formation of cross-joints), makes local sequences very different from "tectonic events". | Thank you for pointing out that these references are not appropriate for backing up this point. We have changed the reference to (e.g. Vitale et al., 2012) to be more appropriate to the point raised. |
| 19 | 493-509 | I do not understand how these Pc values are calculated, and how values for faults and joints rea integrated to discuss the connectivity of the network as a whole.

Fig 9 shows plots for the combined network. This looks perfectly sensible, but who these values are derived is not explained in caption or text. | Pc is calculated separately for the fault, joint, and combined network. The connectivity of the fault network represents the network of mineralised features that display visible offsets. While features may abut against pre-existing phase 1-4 joints (Fig.8), this is not considered as connected with regards to the ‘fault-network’. The connectivity of the fault network is important as the more connected it becomes, the less hydrologically connected the open fractures on the limestone pavement become.

Because the fault-network is mineralised, it is no longer appropriate to consider points |

| | | | |
|---|---|---|---|
| | | | where open-mode fractures abut against faults as being connected (Y-node) and instead they are considered as I-nodes due to the sealing properties of the fault.

This explains why faulting becomes more intense and connected, the joint network (which represents the modern day connectivity of the network) decreases.

To make this clear we have added the following text to the revised MS: “The drop on connected joints is shown in the trends (pink arrows) on Figure 9 and is caused by the gradual increase in abutting relationships between fault’s and joints. As more joints become reactivated as faults, the fault network becomes more connected as splays (i.e. y-nodes; Figure 8) develop, whilst reducing the number of connected joints (i.e. x- and y- nodes in the ‘joint’ dataset) (Figure 8). Similarly, as the intensity and connectivity of the fault network increases, the number of abutting relationships between joints and faults increases. The increases the number of i-nodes in the joint network and gradually decreases the number of connected branches as the intensity of faulting increases.” (Lines 524 to 530), and “see main text for a description of the trends” to the figure caption of Figure 9. |
| 20 | 574-579 | What happened to Stage 1b in Table 4? | We have added this into the conclusion paragraph. “Pre-existing weaknesses developed in the fluvial deltaic sequences at Spireslack SCM as cleats and joints formed during burial of the fluvial-deltaic host rocks and formation of the regional Muirkirk syncline (***Stage 1b***).” please see Lines 595 to 597 of the revised MS. |